# Ensemble-based enzyme design can recapitulate the effects of laboratory directed evolution in silico

Aron Broom[1,4], Rojo V. Rakotoharisoa[1,4], Michael C. Thompson [2,3], Niayesh Zarifi [1], Erin Nguyen[1], Nurzhan Mukhametzhanov[1], Lin Liu [2], James S. Fraser [2] & Roberto A. Chica [1✉]

The creation of artificial enzymes is a key objective of computational protein design. Although de novo enzymes have been successfully designed, these exhibit low catalytic efficiencies, requiring directed evolution to improve activity. Here, we use room-temperature X-ray crystallography to study changes in the conformational ensemble during evolution of the designed Kemp eliminase HG3 ($k_{cat}/K_M$ 146 $M^{-1}s^{-1}$). We observe that catalytic residues are increasingly rigidified, the active site becomes better pre-organized, and its entrance is widened. Based on these observations, we engineer HG4, an efficient biocatalyst ($k_{cat}/K_M$ 103,000 $M^{-1}s^{-1}$) containing key first and second-shell mutations found during evolution. HG4 structures reveal that its active site is pre-organized and rigidified for efficient catalysis. Our results show how directed evolution circumvents challenges inherent to enzyme design by shifting conformational ensembles to favor catalytically-productive sub-states, and suggest improvements to the design methodology that incorporate ensemble modeling of crystallographic data.

[1] Department of Chemistry and Biomolecular Sciences, University of Ottawa, 10 Marie Curie, Ottawa, ON K1N 6N5, Canada. [2] Department of Bioengineering and Therapeutic Science, University of California, San Francisco, San Francisco, CA 94158, USA. [3] Department of Chemistry and Chemical Biology, University of California, Merced, Merced, CA 95343, USA. [4]These authors contributed equally: Aron Broom, Rojo V. Rakotoharisoa. ✉email: rchica@uottawa.ca

Enzymes are the most efficient catalysts known, accelerating chemical reactions by up to 26 orders of magnitude[1] while displaying unmatched selectivity. The ability to create, from scratch, an efficient artificial enzyme for any desired chemical reaction (i.e., a de novo enzyme) is a key objective of computational protein design. Progress towards this goal has been made over the past few decades following the development of computational enzyme design algorithms[2,3]. These methods have been used to create de novo enzymes for a variety of model organic transformations including the Kemp elimination[4,5], retro-aldol[6,7], Diels-Alder[8], ester hydrolysis[9], and Morita-Baylis-Hilman[10] reactions. Although successful, catalytic activities of de novo enzymes have been modest, with $k_{cat}/K_M$ values being several orders of magnitude lower than those of natural enzymes[11,12]. In addition, structural analyses of designed enzymes have revealed important deficiencies in the computational methodologies, resulting in inaccurate predictions of catalytic and ligand-binding interactions[5], and thereby low success rates[4,6,8], emphasizing the need for the continued development of robust enzyme design algorithms.

To improve the catalytic activity of designed enzymes, researchers have used directed evolution. This process has yielded artificial enzymes displaying catalytic efficiencies approaching those of their natural counterparts and provided valuable information about the structural determinants of efficient catalysis[4,13–15]. During evolution, active-site residues, including designed catalytic amino acids, were often mutated, leading to enhanced catalysis via the introduction of new catalytic groups, optimization of catalytic contacts and ligand-binding modes, and enhanced transition-state complementarity of the binding pocket[13–15]. Directed evolution has also yielded beneficial mutations at positions remote from the active site. Distal mutations have been shown to enhance catalysis by shifting the populations of conformational sub-states that enzymes sample on their energy landscape towards those that are more catalytically active[16–18]. Therefore, a better understanding of enzyme conformational ensembles, including the effect of mutations on the population of sub-states, could provide valuable insights to aid in

the development of robust computational enzyme design methodologies.

Here, we study changes in the conformational ensemble along the evolutionary trajectory of the de novo Kemp eliminase HG3 ($k_{cat}/K_M$ 146 $M^{-1} s^{-1}$) using room-temperature X-ray crystallography. We observe that during evolution, catalytic residues were increasingly rigidified through improved packing, the active site became better pre-organized to favor productive binding of the substrate, and the active-site entrance was widened to facilitate substrate entry and product release. Based on these observations, we generate a variant that contains all mutations necessary to establish these structural features, which are found at positions within or close to the active site. This variant, HG4, is >700-fold more active than HG3, with a catalytic efficiency on par with that of the average natural enzyme ($k_{cat}/K_M$ 103,000 $M^{-1} s^{-1}$). Crystallographic analysis of HG4 reveals that mutations proximal to the active site are sufficient to alter the conformational ensemble for the enrichment of catalytically competent sub-states. Lastly, we demonstrate that HG4 can be successfully designed using a crystallographically derived ensemble of backbone templates approximating conformational flexibility, but not with the single template used to design HG3, offering insights for improving enzyme design methodologies.

## Results

**HG series of Kemp eliminases.** Perhaps the most successful example of the improvement of a de novo enzyme by directed evolution has been the engineering of HG3.17, the most active Kemp eliminase reported to date[11]. This artificial enzyme catalyzes the concerted deprotonation and ring-opening of 5-nitrobenzisoxazole into the corresponding o-cyanophenolate (Fig. 1a) with a reported catalytic efficiency of $2.3 \times 10^5 M^{-1} s^{-1}$ [15]. HG3.17 was evolved from HG3, a higher-activity mutant (S265T) of the in silico design HG2 (Supplementary Fig. 1) that was engineered post-design to reduce the active-site conformational heterogeneity observed by molecular dynamics analysis of HG2[5]. Over an evolutionary trajectory that yielded the HG3.3b, HG3.7, and HG3.14 intermediates (Fig. 1b, Supplementary Table 1), a total of 17

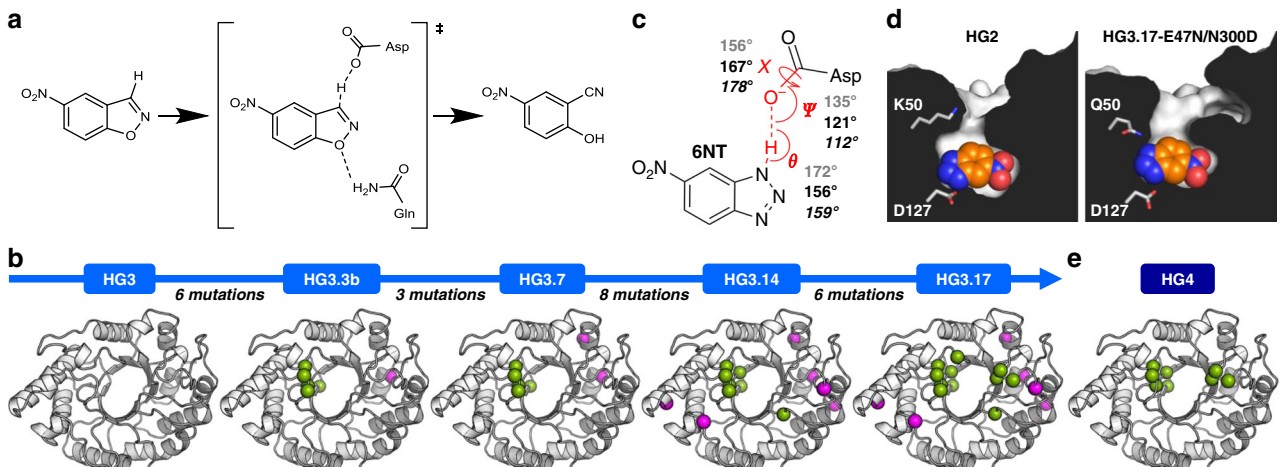

**Fig. 1 HG series of Kemp eliminases. a** HG enzymes catalyze the Kemp elimination reaction using a catalytic dyad consisting of a base (Asp127) that deprotonates 5-nitrobenzisoxazole, and an H-bond donor (Gln50) that stabilizes negative charge buildup on the phenolic oxygen at the transition state (‡). This reaction yields 4-nitro-2-cyanophenol. **b** Directed evolution of HG3, a higher-activity mutant (S265T) of the in silico design HG2. A total of 17 mutations (shown as spheres) were introduced during evolution, including 11 at positions within or close to the active site (green) and 6 at distal sites (magenta). **c** Angles describing the hydrogen bonding interaction between the transition-state analogue 6-nitrobenzotriazole (6NT) and Asp127 in the HG2 (PDB ID: 3NYD)[5] and HG3.17-E47N/N300D (PDB ID: 4BS0)[15] crystal structures are indicated in gray and black, respectively. Values in italics are optimal angles calculated for hydrogen bonding interactions between acetamide dimers[54]. **d** Cut-away view of the active-site pocket shows that its structural complementarity with 6NT (spheres) is improved in the higher-activity variant HG3.17-E47N/N300D. Key active-site residues are shown as sticks. **e** The HG4 variant engineered in this study contains 8 first- and second-shell mutations found during evolution.

**Table 1 Kinetic parameters of Kemp eliminases.**

| Enzyme | Mutations from HG3[a] | $k_{cat}/K_M$ (M$^{-1}$ s$^{-1}$)[b] |
|---|---|---|
| HG3 | – | 146 ± 6 (1300) |
| HG3.3b | V6I *K50H M84C* S89R *Q90D A125N* | 2200 ± 100 (5400) |
| HG3.7 | V6I *Q37K K50Q M84C* S89R *Q90H A125N* | 27,000 ± 2000 (37,000) |
| HG3.14 | V6I *Q37K K50Q* G82A *M84C Q90H* T105I *A125T* T142N T208M T279S D300N | 52,000 ± 1000 (70,000) |
| HG3.17 | V6I *Q37K* N47E *K50Q* G82A *M84C* S89N *Q90F* T105I *A125T* T142N T208M *F267M* W275A R276F T279S D300N | 126,000 ± 9000 (230,000) |
| HG4 | *K50Q* G82A *M84C Q90F A125T F267M* W275A R276F | 103,000 ± 4000 |

[a]Mutations in italics occurred at sites optimized during the computational design of HG2[5].
[b]Individual parameters $K_M$ and $k_{cat}$ could not be determined accurately because saturation was not possible at the maximum substrate concentration tested (2 mM), which is the substrate's solubility limit (Supplementary Fig. 2). Catalytic efficiencies ($k_{cat}/K_M$) were calculated from the slope of the linear portion ([S] ≪ $K_M$) of the Michaelis–Menten model ($v_0 = (k_{cat}/K_M)[E_0][S]$). $n = 2$ independent experiments for HG3, HG3.3b, HG3.7, and HG3.17. $n = 3$ independent experiments for HG3.14 and HG4. Errors of linear regression fitting, which represent the absolute measure of the typical distance that each data point falls from the regression line, are provided. Values in parentheses are from Blomberg et al.[15].

mutations were introduced into HG3 to produce HG3.17, resulting in a catalytic efficiency increase of approximately three orders of magnitude (Table 1, Supplementary Fig. 2). Of these mutations, 11 occurred at positions within or close to the active site, including 8 at positions that were optimized during the computational design of HG2 (Table 1). One of the key active-site mutations occurred at position 50, which was mutated twice during evolution, first from lysine to histidine (HG3 to HG3.3b) and then from histidine to glutamine (HG3.3b to HG3.7), resulting in a novel catalytic residue ideally positioned for stabilizing negative charge buildup on the phenolic oxygen at the transition state (Fig. 1a). Comparison of the crystal structure of the in silico design HG2 (PDB ID: 3NYD)[5] with that of a double mutant of HG3.17, in which surface mutations N47E and D300N were reverted to the corresponding amino acids found in HG2 to facilitate crystallization (HG3.17-E47N/N300D, PDB ID: 4BS0)[15], revealed that catalytic activity was also enhanced via optimized alignment of the transition-state analogue 6-nitrobenzotriazole (6NT) with the catalytic base Asp127 (Fig. 1c), and improved active-site complementarity to this ligand (Fig. 1d). Given that subtle changes to the conformational ensemble of an enzyme can lead to significant rate enhancements[16–18], it is possible that mutations in HG3.17 also contributed to enhanced catalytic efficiency by altering the conformational landscape to enrich catalytically competent sub-states. However, the structures of HG2 and HG3.17-E47N/N300D were solved in the presence of bound 6NT and at cryogenic temperatures, which could have shifted the conformational ensemble towards a single predominant sub-state, thereby limiting our ability to evaluate changes to the conformational landscape during directed evolution.

**Room-temperature crystal structures.** To evaluate changes to the HG3 conformational ensemble along its evolutionary trajectory, we solved room-temperature (277 K) X-ray crystal structures of all HG-series Kemp eliminases, both in the presence and absence of bound 6NT. Room-temperature X-ray crystallography can reveal conformational heterogeneity in protein structures that would not be visible at cryogenic temperatures and thereby provide insights into the conformational ensemble that is sampled by a protein in solution[19]. All five enzymes yielded crystals under similar conditions (Supplementary Table 2), and these diffracted at resolutions of 1.35–1.99 Å (Supplementary Table 3). All unit cells corresponded to space group $P2_12_12_1$ with two protein molecules in the asymmetric unit, except that of HG3.17, whose asymmetric unit was half the volume of the others and contained only one polypeptide chain although the space group was also $P2_12_12_1$. This result is in contrast with the deposited structure of HG3.17-E47N/N300D, which contains two molecules in the asymmetric unit, with identical space groups and similar unit cell dimensions to those of all other HG variants reported

here[15]. This discrepancy between our structure of HG3.17 and the previously published structure of HG3.17-E47N/N300D is likely caused by the presence of the Asn47 surface residue in all variants except for HG3.17, since this amino acid is involved in crystal packing interactions.

All HG-series enzymes bound 6NT in the same catalytically productive pose (Fig. 2a) as that observed in HG2 and HG3.17-E47N/N300D (Fig. 1d). In this pose, the acidic N–H bond of 6NT that mimics the cleavable C–H bond of the substrate is located within hydrogen-bonding distance to the carboxylate oxygen of Asp127 (2.5–2.6 Å distance between heavy atoms), while the basic nitrogen atom corresponding to the phenolic oxygen of the transition state forms an H-bond with either a water molecule (HG3), the $N_\varepsilon$ atom of His50 (HG3.3b), or the side-chain amide nitrogen of Gln50 (HG3.7, HG3.14, HG3.17). In addition to being held in place by these polar interactions, 6NT is sandwiched between the hydrophobic side chains of Trp44 and Met237 (Fig. 2b), which are part of a mostly hydrophobic binding pocket that also includes the side chains of Ala21, Met/Cys84, Met172, Leu236, Thr265, and Phe/Met267, as well as the backbone of Gly83 and Pro45 (Supplementary Fig. 3). Interestingly, the *cis* peptide bond formed between residues 83 and 84 that is present in the *Thermoascus aurantiacus* xylanase 10 A structure used as template for computational design (PDB ID: 1GOR[20]) is maintained in all HG structures (Fig. 2c) even though both residues were mutated to obtain HG3 (H83G and T84M). In addition to adopting a *cis* conformation, which is stabilized by hydrogen bonding to an ordered water molecule, this peptide bond also adopts the *trans* conformation in the structures of 6NT-bound HG3 and HG3.3b (Fig. 2c, Supplementary Fig. 4). However, starting at HG3.7, the peptide bond is found exclusively in the *cis* conformation in the 6NT-bound structures because it is stabilized by an additional hydrogen bond with the Gln50 side-chain carbonyl oxygen. This hydrogen bonding interaction helps to lock Gln50 in a conformation that is properly oriented to stabilize negative charge buildup on the phenolic oxygen at the transition state. The introduction of this new catalytic group in a catalytically productive conformation likely accounts for the majority of the 12-fold enhancement in $k_{cat}/K_M$ observed in HG3.7 relative to HG3.3b, a hypothesis that is supported by the 16-fold decrease in $k_{cat}/K_M$ observed when the Q50H mutation is introduced into HG3.17[21].

From HG3.7 to HG3.17, no further changes in catalytic residues occurred during evolution. Yet, catalytic efficiency increased by approximately fivefold (Table 1). To evaluate whether this increase in activity was caused by changes to the conformational ensemble, we analyzed the B-factors of catalytic residues, which can be interpreted as a measure of the average displacement of an atom, or group of atoms, in the crystal. Since both conformational heterogeneity and crystalline disorder can

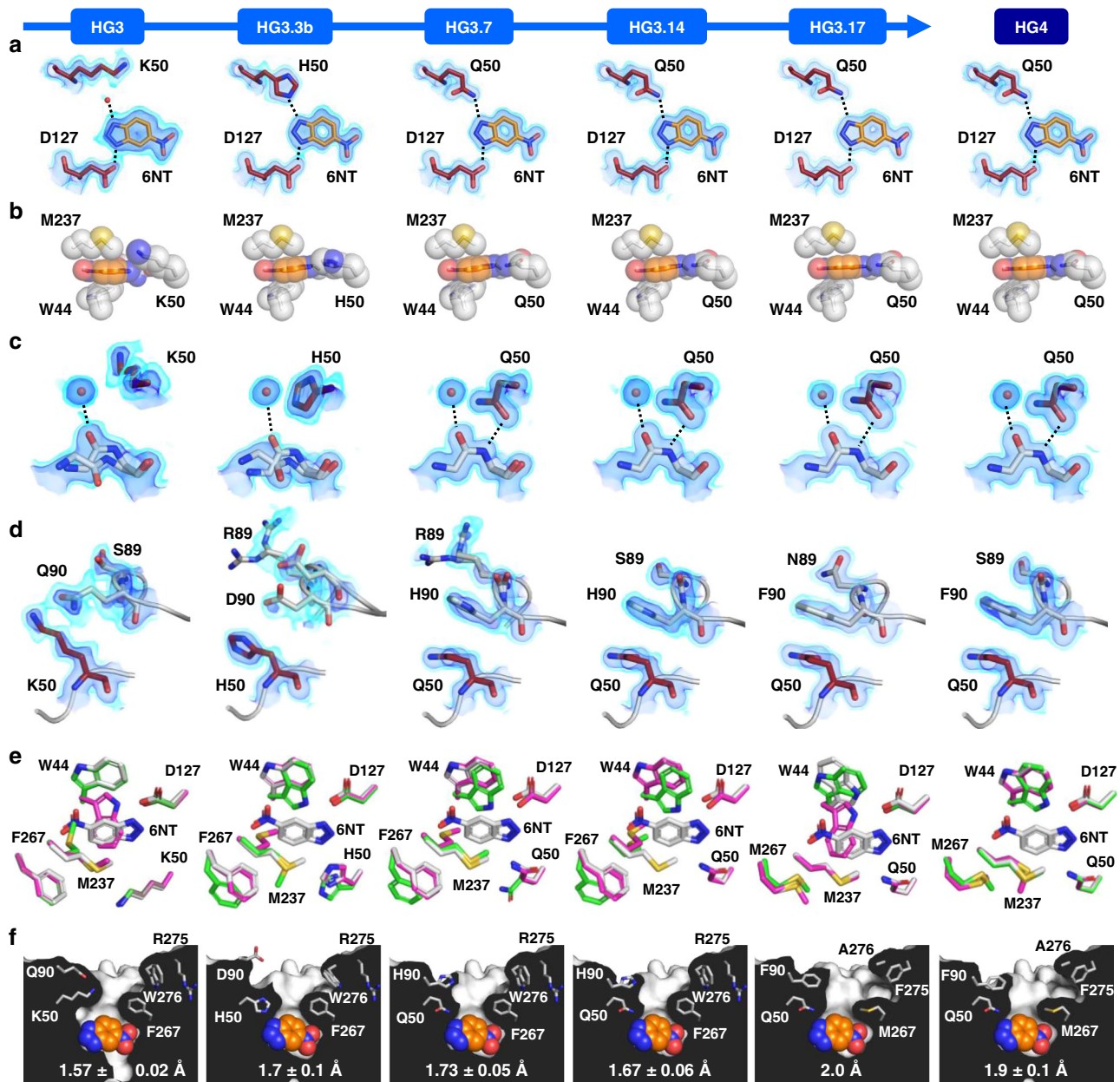

**Fig. 2 Crystal structures of HG-series Kemp eliminases.** In all cases, only atoms from chain A are shown. **a** Binding pose of the 6-nitrobenzotriazole (6NT) transition-state analogue (orange). Hydrogen bonds are shown as dashed lines. The red sphere represents a water molecule. The 2Fo-Fc map is shown in volume representation at two contour levels: 0.5 and 1.5 eÅ$^{-3}$ in light and dark blue, respectively. **b** 6NT (orange) is sandwiched between the hydrophobic side chains of Trp44 and Met237. **c** The peptide bond between residues 83 and 84 can adopt *cis* or *trans* conformations. Hydrogen bonds are shown as dashed lines. The 2Fo-Fc map is shown in volume representation at two contour levels: 0.5 and 1.5 eÅ$^{-3}$ in light and dark blue, respectively. **d** Conformational changes to the loop formed by residues 87–90 over the course of the evolutionary trajectory. The 2Fo-Fc map is shown in volume representation at two contour levels: 0.5 and 1.5 eÅ$^{-3}$ in light and dark blue, respectively. **e** Superposition of the 6NT-bound structure (white) with the highest (magenta) and lowest (green) occupancy conformers of the unbound structure for each Kemp eliminase. From HG3 to HG3.14, the unbound state is never pre-organized for catalysis as both Trp44 and Met237 adopt conformations that would prevent the productive binding of the transition state. In HG3.17 and HG4 however, only Trp44 adopts a non-productive conformation in the unbound state, with an occupancy of 62% or 26%, respectively. **f** Cut-away view of the active site shows that its entrance (top) becomes widened during evolution, as indicated by an increasing bottleneck radius (reported as the average radius ± s.d. calculated using the highest occupancy conformers from both chain A and B, except for HG3.17, which contains a single chain). 6NT is shown as orange spheres. Bottleneck radii were calculated using the PyMOL plugin Caver 3.0[22].

contribute to atomic B-factors, with the latter effect potentially varying between different crystals, we calculated the Z-scores of the atomic B-factors and compared those across our crystal structures of different HG variants. This Z-score analysis allowed us to evaluate the variation of B-factors relative to the mean value within an individual crystal and showed that rigidity of the

Asp127 side chain did not vary significantly during evolution (Fig. 3a). By contrast, the side chain of residue 50 became increasingly rigidified over the course of the evolutionary trajectory. Increasing rigidity at position 50 is expected when this residue is mutated from a lysine to a histidine (HG3 to HG3.3b), given the lower number of degrees of freedom in the

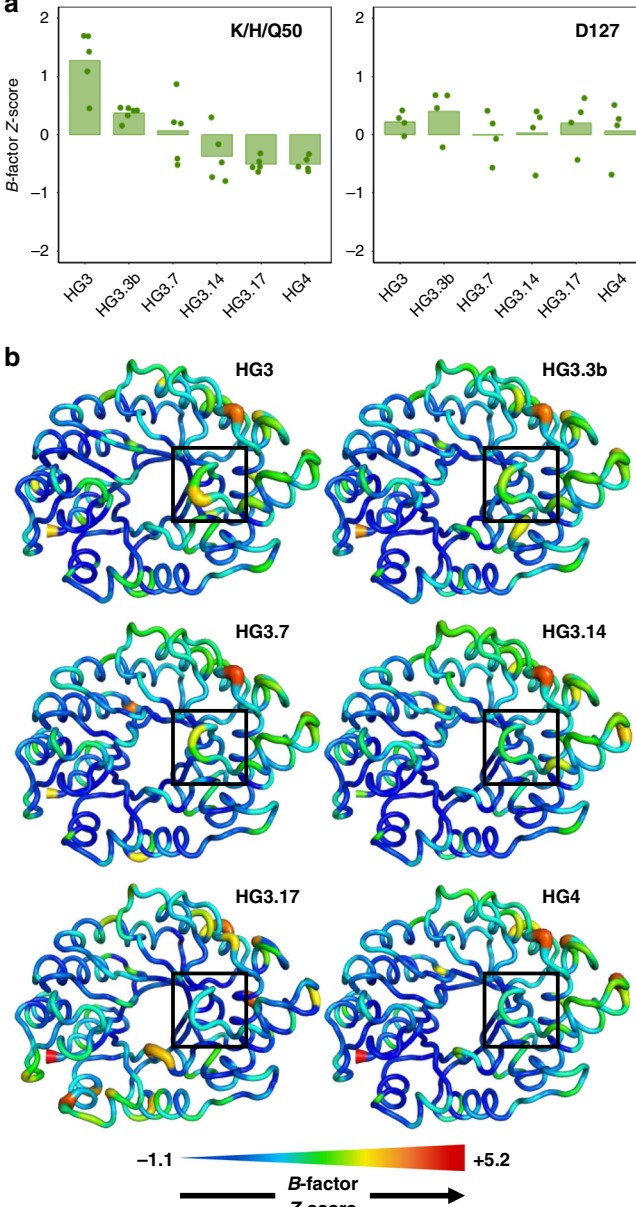

**Fig. 3 Conformational heterogeneity. a** B-factor Z-scores for the residue at position 50 in the absence of bound 6-nitrobenzotriazole (6NT) decrease over the course of the evolutionary trajectory, while those for Asp127 do not change significantly. Z-scores of individual side-chain heavy atoms are shown as dots (values averaged over both chain A and B for all structures except that of HG3.17, which contains a single chain in the asymmetric unit), while the average Z-score for the whole side chain is indicated by the bar. Positive and negative Z-scores indicate increased flexibility or rigidity relative to the average residue in the protein, respectively. **b** B-factor Z-scores of each protein residue (average of all side-chain heavy atoms) in the absence of bound 6NT plotted on a model backbone for each Kemp eliminase. Thickness of the sausage plot increases with the B-factor Z-score, indicating increased flexibility. The loop formed by residues 87–90 (boxed) becomes more rigid during evolution. Source data are provided as a source data file.

latter amino acid. This trend is also expected when histidine at position 50 is mutated to a glutamine (HG3.3b to HG3.7) given the ability of glutamine but not histidine to hydrogen-bond with the *cis* peptide formed by residues Gly83 and Cys84 (Fig. 2c). However, rigidity continues to increase at this position between

HG3.7 and HG3.17, even though the side-chain rotamer of Gln50 in the presence of bound 6NT remains the same (Fig. 2a). This result suggests that other structural features contribute to the increased rigidity observed at this position.

To verify the underlying cause of the increased rigidity at position 50, we calculated the average Z-score of atomic B-factors for each residue. We observed a trend whereby the loop formed by residues 87–90, which is located directly on top of residue 50, becomes increasingly rigidified during evolution (Fig. 3b). Interestingly, two residues forming this loop (89 and 90) were mutated multiple times over the course of the evolutionary trajectory (Table 1). These mutations induce a conformational change in the loop that moves it closer to the active site, which results in a pi-stacking interaction between the phenyl and carboxamide groups of Phe90 and Gln50 that increases the rigidity of the catalytic residue (Fig. 2d, Supplementary Fig. 5a).

Although increasing rigidity of the Gln50 catalytic residue from HG3.7 to HG3.17 likely contributes to enhanced catalysis, other structural effects were investigated. A key determinant of efficient enzyme catalysis is active site pre-organization, which enables enzymes to bind substrates in a geometry close to that of the transition state. To evaluate changes in active site pre-organization during evolution, we compared the structures of HG-series Kemp eliminases in the presence and absence of bound 6NT. In all enzymes except for HG3.17, the unbound state is never pre-organized for catalysis as both Trp44 and Met237 adopt conformations that would prevent productive binding of 6NT (Fig. 2e). In addition, the His50 and Gln50 catalytic residues in HG3.3b and HG3.7, respectively, adopt a low-occupancy, catalytically non-productive conformation in the unbound state that cannot interact favorably with 6NT. Interestingly, the non-productive conformation of Gln50 in the HG3.7 unbound state (26% occupancy) cannot stabilize the *cis* peptide bond formed by residues 83 and 84 via a hydrogen bonding interaction, and accordingly, the *trans* peptide conformation is also observed in this structure (25% occupancy) (Supplementary Figs. 4 and 5b).

In contrast with all other HG variants, the unbound state of HG3.17 is correctly pre-organized for catalysis in a large portion of the molecules in the crystal, with only Trp44 adopting a non-productive conformation at 62% occupancy (Fig. 2e). In this variant, Met237 adopts exclusively the productive conformer in the unbound state, which is stabilized by packing interactions with the neighboring Met267 side chain, a mutation that was introduced late in the evolutionary trajectory (HG3.14 to HG3.17). Overall, three of the four residues that are key for binding and stabilizing 6NT (Gln50, Asp127, Met237) adopt a catalytically productive conformation in the HG3.17 unbound state, resulting in ~40% of the molecules in the crystal being correctly pre-organized for efficient catalysis.

Enhanced complementarity to the transition state is another important feature of efficient catalysis. Therefore, computational enzyme design algorithms aim to optimize the packing of the transition state. However, transition-state overpacking may reduce catalytic efficiency by creating a high-energy barrier preventing substrate entry and product release. To evaluate whether active-site accessibility changed during evolution, we calculated the active-site entrance bottleneck radius on 6NT-bound structures[22]. We observed that during evolution, the active-site bottleneck formed by the side chains of residues 50 and 267, became widened (Fig. 2f), as did the mouth of the substrate entry channel formed by residues Arg275 and Trp276, which were mutated to smaller amino acids. This widening of the active site entrance could help to eliminate high-energy barriers to substrate entry and product release that could have been caused by tighter packing of 6NT in higher-activity HG variants.

**HG4 is an efficient artificial enzyme.** All of the structural features that enhance the activity described above are caused primarily by residues within or close to the active site, which suggests that mutagenesis far from the active site may not be essential to create an efficient artificial enzyme. To test this hypothesis, we generated a variant of HG3 that contains all HG3.17 mutations found within 7.5 Å of 6NT, with the exception of N47E, which we omitted to favor the formation of a unit cell similar to that of HG3. We also included the second-shell W275A and R276F mutations found to widen the active site entrance. This yielded HG4, a variant of HG3 containing 8 mutations (Fig. 1e, Supplementary Table 1). Kinetic analysis of HG4 revealed that its catalytic efficiency is >700-fold higher than that of HG3 (Table 1, Supplementary Fig. 2), and equivalent to that of the average natural enzyme ($\sim 10^5\,M^{-1}\,s^{-1}$)[23]. Crystallographic analysis of HG4 (Supplementary Tables 2–3) showed that its structure is highly similar to that of HG3.17 but with an active site that is better pre-organized (Figs. 2, 3, Supplementary Figs. 3 and 5). However, HG4 is ~20% less active than HG3.17, demonstrating that the additional 9 mutations found in the latter enzyme, most of which are distal to the active site, play a role in enhancing catalytic efficiency.

**Computational design of HG4.** Given that all but one mutation (G82A) in HG4 are found at sites that were optimized during the design of HG2[5], we investigated whether the HG4 structure could be accurately predicted using a computational protocol similar to the one that produced HG2 ("Methods", Supplementary Tables 4–6). To do so, we first performed a positive control calculation in which rotamers for the HG4 sequence were optimized on the crystal structure backbone of 6NT-bound HG4. This calculation yielded an in silico model of HG4 with an energy score and a predicted rotameric configuration in excellent agreement with the crystal structure (Fig. 4a). This control demonstrates that the combination of the energy function, rotamer library, and search algorithm used in this protocol is sufficiently accurate for recapitulating the structure of HG4, provided that the correct template, binding pose, and catalytic dyad are allowed. By contrast, when we replaced the HG4 backbone template with the *Thermoascus aurantiacus* xylanase 10 A backbone used to design HG2 (PDB ID: 1GOR)[20], we obtained a structural model that differs significantly from the HG4 crystal structure and that is destabilized by ~45 kcal/mol (Fig. 4b). This result demonstrates that the 1GOR backbone template is not well-suited to accommodate the HG4 sequence, as evidenced by differences between the 1GOR-derived model and the HG4 crystal structure. Specifically, the backbone at position 83 is shifted by 1.1 Å in the HG4 crystal structure relative to its position in the 1GOR template, causing the transition state to adopt an alternate binding pose that minimizes steric clashes with Gly83, which is accompanied by repacking of several residues around the transition state, including Gln50. Use of our HG3 crystal structures with or without 6NT as the design template causes similar, but less severe, structural, and energetic effects (Fig. 4c, d). These results highlight the impact of small backbone geometry variations on predictions made by enzyme design.

To address issues arising from the use of a single fixed backbone template, we generated backbone ensembles using molecular dynamics restrained by the HG3 or 1GOR diffraction data, also known as ensemble refinement (Methods), and used the resulting templates to optimize rotamers for the HG4 sequence. We were able to recapitulate the correct transition-state binding mode on several individual ensemble members derived from the HG3 crystallographic data, with energies comparable to that of the HG4

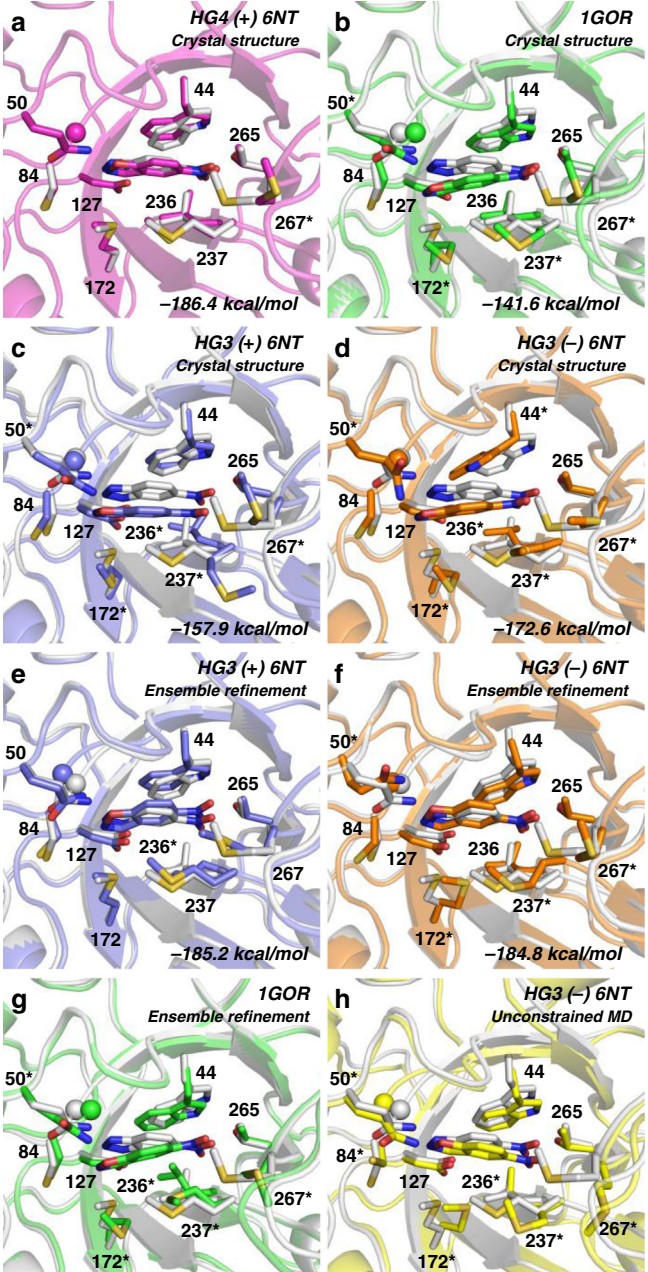

**Fig. 4 Computational design of HG4 on various backbone templates.** The HG4 crystal structure with bound 6-nitrobenzotriazole (white) is overlaid on the HG4 design models (colored) obtained using the crystal structure of (**a**) HG4 with bound 6-nitrobenzotriazole, **b** *Thermoascus aurantiacus* xylanase 10 A (PDB ID: 1GOR), **c** HG3 with bound 6-nitrobenzotriazole, or (**d**) HG3 without 6-nitrobenzotriazole. **e–h** the HG4 design models obtained using the template prepared by ensemble refinement or unconstrained molecular dynamics (MD) that gave the best energy following repacking. PHOENIX energies of design models after repacking are indicated at the bottom right. For reference, the energy of the HG4 crystal structure with a bound transition state is −186.7 kcal/mol. In all cases, the transition state and transition-state analogue are shown at the center of the barrel. Side chains of all residues forming the binding pocket are shown with the exception of Ala21 and Pro45, which were omitted for clarity. The sphere shows the alpha carbon of Gly83. Asterisks indicate residues that adopt side-chain rotamers varying by >20 degrees around one or more side-chain dihedrals between the design model and crystal structure.

crystal structure (Fig. 4e, f, Supplementary Fig. 6). However, the use of an ensemble derived from the 1GOR diffraction data did not allow recapitulation of the crystallographic transition-state binding mode (Fig. 4g) although it did yield several computational models displaying improved energy (Supplementary Fig. 6). The inferior performance of the 1GOR-derived ensemble compared to the HG3 ensembles likely results from differences in conformational heterogeneity within the ensemble, specifically at position 83 (Supplementary Fig. 7). To evaluate the effect of restraints imposed by the diffraction data, we generated an ensemble using unconstrained molecular dynamics starting from the unbound HG3 crystal structure (Methods), and used it to optimize rotamers for the HG4 sequence. The use of this ensemble resulted in an improved structural model (Fig. 4h) compared to the one obtained from the corresponding crystal structure (Fig. 4d) that is however less structurally accurate and stable than the one obtained from the ensemble refinement (Fig. 4f). The better predictive ability of the HG3-derived ensembles prepared using crystallographic restraints likely results from their lower deviation from the HG4 crystal structure (≈0.4 Å, Supplementary Fig. 7), which we previously showed to be necessary for an ensemble to represent a physically valid model of the target protein fold[24]. Overall, these results suggest that computational enzyme design with a crystallographically derived backbone ensemble derived from a low-activity enzyme could obviate the need for directed evolution by allowing catalytically competent sub-states to be sampled during the design procedure.

## Discussion

In this work, we followed changes to the conformational ensemble that occur during the evolution of an enzyme with de novo biocatalytic function. Unlike previous examples where the active sites of de novo enzymes were completely remodeled during evolution[25,26], or where the binding pose of the substrate or transition-state analogue was significantly altered[13,17], we observed only subtle changes to the active site geometry or 6NT-binding pose in the HG-series of Kemp eliminases. By contrast, many of the structural changes that contribute to enhanced catalysis in the HG series are dynamic in nature: the Gln50 catalytic residue became more rigid even though its average structure did not vary substantially, and the active site became better pre-organized via enrichment of catalytically productive conformations of 6NT-binding residues that were already present in the unbound state. These observations illustrate how small changes to the active site conformational ensemble can drive large changes in catalytic efficiency. Since these changes can be subtle and difficult to predict computationally, directed evolution can help increase activity by selecting for mutations that enrich catalytically competent sub-states[17,18].

Despite the challenges inherent to enzyme design, which are highlighted by our observations of the effects of mutations in the HG series of Kemp eliminases, our results suggest that de novo enzymes with native-like catalytic efficiencies can be computationally designed, without the need to rely on subsequent improvement by laboratory directed evolution. Indeed, all mutations found in HG4 relative to the wild-type *Thermoascus aurantiacus* xylanase 10 A template from which it is derived (PDB ID: 1GOR) are found at either first or second-shell residues, and these sites were all optimized during the original design of HG2[5]. Yet, Privett et al. designed the lower activity enzyme HG2 instead of HG4. While Gln50 was not sampled as part of the catalytic dyad during the design of HG2, the combination of the Asp127/Gln50 dyad with the productive transition-state binding pose would have scored poorly on the 1GOR template regardless. However, our approach to computational enzyme design that

utilized an experimentally derived ensemble of backbone templates yielded HG4 models with energies and binding modes comparable to that of the HG4 crystal structure. These results suggest an iterative approach to computational enzyme design that could circumvent the need for directed evolution by introducing an additional round of design that utilizes a backbone ensemble generated from experimental structural data obtained for an initial, low-activity enzyme. In the case of evolution, mutations are not selected for in the context of a single backbone conformation but instead across an entire conformational ensemble[18]. Our ensemble design approach should therefore be more accurate than traditional approaches relying on a single backbone template because it allows the accessible conformational ensemble to be represented in the scoring of sequences. The incorporation of experimental restraints in the generation of the ensemble ensures that the computational procedure is applied to the true conformational ensemble that is sampled by the enzyme.

The results reported here provide additional support for the well-known fact that enzymes are plastic molecules whose backbone conformation can change upon introduction of mutations (as seen when comparing the 1GOR and HG-series crystal structures), and suggest improvements to the enzyme design protocol that can account for this property. This could be achieved by incorporating flexible backbone design algorithms during the repacking step[27,28], or by using pre-generated ensembles of energetically accessible backbone templates[24,29], as was done here. While these methodological changes may improve the design of the enzyme transition state, it is likely that the creation of de novo enzymes with native-like catalytic efficiencies for more complex reactions will require a holistic approach where every possible state that the enzyme samples along its reaction coordinate is included in the design calculation. This could be achieved by the implementation of multistate approaches to computational protein design that allow the design of protein-energy landscapes[30], rather than single structures. We expect that the structures reported here, especially those of HG4 and HG3, will be helpful to benchmark these future enzyme design protocols.

## Methods

**Protein expression and purification.** Codon-optimized and his-tagged (C-terminus) genes for HG-series Kemp eliminases (Supplementary Table 7) cloned into the pET-11a vector (Novagen) via *Nde*I and *Bam*HI were obtained from Genscript. Enzymes were expressed in *E. coli* BL21-Gold (DE3) cells (Agilent) using lysogeny broth (LB) supplemented with 100 μg mL$^{-1}$ ampicillin. Cultures were grown at 37 °C with shaking to an optical density at 600 nm of 0.3, at which point the incubation temperature was reduced to 18 °C. At an OD600 of 0.6, protein expression was initiated with 1 mM isopropyl β-D-1-thiogalactopyranoside. Following incubation for 16 h at 18 °C with shaking (250 rpm), cells were harvested by centrifugation, resuspended in 10 mL lysis buffer (5 mM imidazole in 100 mM potassium phosphate buffer, pH 8.0), and lysed with an EmulsiFlex-B15 cell disruptor (Avestin). Proteins were purified by immobilized metal affinity chromatography using Ni–NTA agarose (Qiagen) pre-equilibrated with lysis buffer in individual Econo-Pac gravity-flow columns (Bio-Rad). Columns were washed twice, first with 10 mM imidazole in 100 mM potassium phosphate buffer (pH 8.0), and then with the same buffer containing 20 mM imidazole. Bound proteins were eluted with 250 mM imidazole in 100 mM potassium phosphate buffer (pH 8.0), and exchanged into 100 mM sodium phosphate buffer (pH 7.0) supplemented with 100 mM sodium chloride using Econo-Pac 10DG desalting pre-packed gravity-flow columns (Bio-Rad). Proteins were further subjected to gel filtration in 50 mM sodium citrate buffer (pH 5.5) and 150 mM sodium chloride using an ENrich SEC 650 size-exclusion chromatography column (Bio-Rad). Purified samples were concentrated using Amicon Ultracel-10K centrifugal filter units (EMD Millipore), and quantified by measuring the absorbance at 280 nm and applying Beer-Lambert's law using calculated extinction coefficients obtained from the ExPASy ProtParam tool (https://web.expasy.org/protparam/).

**Steady-state kinetics.** All assays were carried out at 27 °C in 100 mM sodium phosphate buffer (pH 7.0) supplemented with 100 mM sodium chloride. Triplicate 200-μL reactions with varying concentrations of freshly prepared 5-nitrobenzisoxazole (AstaTech) dissolved in methanol (10% final concentration, pH

of reaction mixture adjusted to 7.0 after addition of methanol-solubilized substrate) were initiated by the addition of ~2 μM HG3, 50 nM HG3.3b, 10 nM HG3.7/HG3.14, or 5 nM HG3.17/HG4. Product formation was monitored spectrophotometrically at 380 nm ($\varepsilon = 15{,}800$ M$^{-1}$ cm$^{-1}$)[5] in individual wells of 96-well plates (Greiner Bio-One) using a SpectraMax 384Plus plate reader (Molecular Devices). Path lengths for each well were calculated ratiometrically using the difference in absorbance of 100 mM sodium phosphate buffer (pH 7.0) supplemented with 100 mM sodium chloride and 10% methanol at 900 and 975 nm (27 °C)[31]. Linear phases of the kinetic traces were used to measure initial reaction rates. Data were fitted to the linear portion of the Michaelis-Menten model ($v_0 = (k_{cat}/K_M)[E_0]$ [S]), and $k_{cat}/K_M$ was deduced from the slope.

**Crystallization**. Enzyme variants were prepared in 50 mM sodium citrate buffer (pH 5.5) at the concentrations listed in Supplementary Table 2. For samples that were co-crystallized with the transition-state analogue, a 100 mM stock solution of 6NT (AstaTech) was prepared in dimethyl sulfoxide (DMSO) and diluted 20-fold in the enzyme solutions for a final concentration of 5 mM 6NT (5% DMSO). For each enzyme variant, we carried out initial crystallization trials in 15-well hanging drop format using EasyXtal crystallization plates (Qiagen) and a crystallization screen that was designed to explore the chemical space around the crystallization conditions reported by Blomberg et al.[15]. Crystallization drops were prepared by mixing 1 μL of protein solution with 1 μL of the mother liquor, and sealing the drop inside a reservoir containing an additional 500 μL of the mother liquor solution. The mother liquor solutions contained ammonium sulfate as a precipitant in sodium acetate buffer (100 mM), and the specific growth conditions that yielded the crystals used for X-ray data collection are provided in Supplementary Table 2. In some cases, a microseeding protocol was required to obtain high-quality crystals. Microseeds were prepared by vortexing crystals in their mother liquor in the presence of glass beads (0.5 mm), and were subsequently diluted into the mother liquor solutions used to form the crystallization drops.

**X-ray data collection and processing**. Prior to X-ray data collection, crystals were mounted in polyimide loops and sealed using a MicroRT tubing kit (MiTeGen). Single-crystal X-ray diffraction data were collected on beamline 8.3.1 at the Advanced Light Source. The beamline was equipped with a Pilatus3 S 6 M detector, and was operated at a photon energy of 11111 eV. Crystals were maintained at 277 K throughout the course of data collection. Each data set was collected using a total X-ray dose of 200 kGy or less, and covered a 180° wedge of reciprocal space. Multiple data sets were collected for each enzyme variant either from different crystals, or if their size permitted, from unique regions of larger crystals.

X-ray data were processed with the Xia2 0.5.492 program (https://doi.org/10.1107/S0021889809045701), which performed indexing, integration, and scaling with the 20180126 version of XDS and XSCALE[32], followed by merging with Pointless as distributed in CCP4 7.0.053[33]. For each variant, multiple individual data sets were merged to obtain the final set of reduced intensities, and the resolution cutoff was taken where the CC$_{1/2}$ and <I/σI> values for the merged intensities fell to ~0.5 and 1.0, respectively. We determined which individual data sets should be combined by evaluating the overall effects of adding or removing individual data sets on the CC$_{1/2}$ and I/σ in the high-resolution bins of the merged data set. Information regarding data collection and processing is presented in Supplementary Table 3. The reduced diffraction data were analyzed with phenix.xtriage (http://www.ccp4.ac.uk/newsletters/newsletter43/articles/PHZ_RWGK_PDA.pdf) to check for crystal pathologies, and no complications were identified.

**Structure determination**. We obtained initial phase information for calculation of electron density maps by molecular replacement using the program Phaser[34], as implemented in v1.13.2998 of the PHENIX suite[35]. Several different HG-series enzymes were used as molecular replacement search models. All members of the HG-series of enzymes crystallized in the same crystal form, containing two copies of the molecule in the crystallographic asymmetric unit, except for HG3.17, which crystallized with only one molecule in the asymmetric unit. To avoid model bias that could originate from using other members of the HG-series as molecular replacement search models, we applied random coordinate displacements ($\sigma = 0.5$ Å) to the atoms, and performed coordinate refinement against the structure factor data before proceeding to manual model building.

Next, we performed iterative steps of manual model rebuilding followed by refinement of atomic positions, atomic displacement parameters, and occupancies using a translation-libration-screw (TLS) model, a riding hydrogen model, and automatic weight optimization. All model building was performed using Coot 0.8.9.2[36] and refinement steps were performed with phenix.refine within the PHENIX suite (v1.13-2998)[35,37]. Restraints for 6NT were generated using phenix.elbow[38], starting from coordinates available in the Protein Data Bank (PDB ligand ID: 6NT)[39]. Further information regarding model building and refinement, as well as PDB accession codes for the final models, are presented in Supplementary Table 3. Time-averaged ensembles were generated for 1GOR, and HG3 with or without ligand, with phenix.ensemble_refinement implemented in PHENIX v.1.15.2-3472. To prepare the structures for ensemble refinement, low-occupancy conformers were removed, and occupancies adjusted to 100% using phenix.

pdbtools. Hydrogen atoms were then added using phenix.ready_set. This procedure yielded ensembles containing 80, 84, or 50 templates from the 1GOR, HG3 (+) 6NT, or HG3 (−) 6NT crystal structures, respectively.

**Unconstrained molecular dynamics**. All simulations were performed using GROMACS 2019.3 (http://www.gromacs.org) with the AMBER99SB forcefield[40]. Long-range electrostatics (>12 Å) were modeled using the particle mesh Ewald method[41], and the LINCS algorithm[42] was used to treat all bonds as constraints, allowing a time step of 2 fs. Heavy atom coordinates of the major conformer from chain A were extracted from the crystal structure of HG3 in the unbound state (PDB ID: 5RG4). Following coordinate extraction, hydrogen atoms were added using Reduce[43], and the resulting protein molecule was placed in an dodecahedral box with periodic boundary conditions where the distance between the protein surface and the box edges was set to 14 Å. After the addition of explicit TIP3P[44] water molecules, charges on protein atoms were neutralized with Na$^+$ and Cl$^-$ counter-ions at a concentration of 0.15 M. The structure was then energy minimized with the steepest descent method to a target maximum force of 1000 kJ mol$^{-1}$ nm$^{-1}$. The system was equilibrated under an NVT ensemble for 125 ps at a temperature of 300 K using a Nose-Hoover thermostat[45], while applying position restraints for heavy protein atoms. A second equilibration step under an NPT ensemble was performed for 1 ns with a constant pressure and temperature of 1 bar and 300 K, respectively, using the Berendsen barostat[46]. Following removal of the position restraints, a 500-ns production run under Parrinello-Rahman pressure coupling[47] was initiated from the final snapshot of the NPT equilibration. At the end of the simulation, 50 snapshots separated by 10 ns along the production trajectory were extracted. This 50-member ensemble was energy-minimized with a gradient-based tolerance of 0.1 kcal mol$^{-1}$ with the Cartesian minimization application included in the Triad protein design software (Protabit, Pasadena, CA, USA) using the PHOENIX energy function[5] with added covalent terms from the DREIDING forcefield[48].

**Computational enzyme design**. All calculations were performed with the Triad protein design software (Protabit, Pasadena, CA, USA) using a Monte Carlo with simulated annealing search algorithm for rotamer optimization. The crystal structure of *Thermoascus aurantiacus* xylanase 10 A was obtained from the Protein Data Bank (PDB code: 1GOR[20]) and further refined as described above to fix modeling issues with Thr84. Structures of HG3 with and without 6NT, HG4 with 6NT, and ensembles of 1GOR or HG3-derived templates were obtained from the refinement of crystallographic data as described above. Following extraction of protein heavy-atom coordinates for the highest occupancy conformer from chain A, hydrogen atoms were added using the *addH.py* application in Triad. The Kemp elimination transition-state (TS) structure[49] was built with the parameters described by Privett and coworkers[5]. Residue positions surrounding Asp127 were mutated to Gly (Supplementary Table 4), with the exception of position 50, which was mutated to Gln. The 2002 Dunbrack backbone-independent rotamer library[50] with expansions of ±1 standard deviation around $\chi_1$ and $\chi_2$ was used to provide side-chain conformations. A library of TS poses was generated in the active site by targeted ligand placement[2] using the contact geometries listed in Supplementary Table 5. TS pose energies were calculated using the PHOENIX energy function[5], which consists of a Lennard-Jones 12–6 van der Waals term from the Dreiding II force field[48] with atomic radii scaled by 0.9, a direction-dependent hydrogen bond term with a well depth of 8.0 kcal mol$^{-1}$ and an equilibrium donor-acceptor distance of 2.8 Å[51], an electrostatic energy term modeled using Coulomb's law with a distance-dependent dielectric of 10, an occlusion-based solvation potential with scale factors of 0.05 for nonpolar burial, 2.5 for nonpolar exposure, and 1.0 for polar burial[52], and a secondary structural propensity term[53]. During the energy calculation step, TS–side-chain interaction energies were biased to favor interactions that satisfy contact geometries (Supplementary Table 6) as described by Lassila et al.[2].

Following ligand placement, the 10 lowest energy TS poses found on each template (HG4 with 6NT, 1GOR, HG3 with 6NT, and HG3 without 6NT) were selected as starting points for repacking of the HG4 sequence. For individual members of the 1GOR and HG3-derived ensembles, only the single lowest energy TS pose was used for repacking. In the repacking calculation, the TS structure was translated ±0.4 Å in each Cartesian coordinate in 0.2-Å steps, and rotated 10° about all three axes (origin at TS geometric center) in 5° steps for a total combinatorial rotation/translation search size of 5$^6$ or 15,625 poses. Residues that were converted to Gly in the ligand placement step were allowed to sample all conformations of the amino acid found at that position in the HG4 sequence (Supplementary Table 4). The identities of the catalytic residues were fixed and allowed to sample all conformations of that amino-acid type. Side-chain–TS interaction energies were biased to favor those contacts that satisfy the geometries as done during the ligand placement step (Supplementary Table 6). Rotamer optimization was carried out using the search algorithm, rotamer library, and energy function described above. The single lowest energy repacked structure on each backbone template was used for analysis. To compare energies of the HG4 models obtained on the various templates, we calculated the energy difference between each repacked structure and the corresponding all-Gly structure obtained after ligand placement, and these energies are reported throughout the figures and text.

**Statistics and reproducibility**. Experiments were repeated in triplicate where feasible. All replications were successful and the resulting data are presented with error values as described in text. No data was excluded from analyses.

**Reporting summary**. Further information on research design is available in the Nature Research Reporting Summary linked to this article.

## Data availability

Structure coordinates for all HG-series Kemp eliminases have been deposited in the RCSB Protein Data Bank with the following accession codes: HG3 (5RG4, 5RGA), HG3.3b (5RG5, 5RGB), HG3.7 (5RG6, 5RGC), HG3.14 (5RG7, 5RGD), HG3.17 (5RG8, 5RGE), HG4 (5RG9, 5RGF). Source data are provided with this paper. Other relevant data are available from the corresponding author upon reasonable request.

## Code availability

Triad scripts are provided with this paper. The Triad protein design software is available at www.protabit.com.

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

## Acknowledgements

R.A.C. acknowledges an Early Researcher Award from the Ontario Ministry of Economic Development & Innovation (ER14-10-139), and grants from the Natural Sciences and Engineering Research Council of Canada (NSERC, RGPIN-2016-04831) and the Canada Foundation for Innovation (26503). J.S.F. is supported by a Searle Scholar Award from the Kinship Foundation, a Pew Scholar Award from the Pew Charitable Trusts, a Packard Fellowship from the David and Lucile Packard Foundation, NIH GM110580, UC Office of the President Laboratory Fees Research Program LFR-17-476732, and NSF STC-1231306. A.B. is the recipient of a postdoctoral fellowship from NSERC. M.C.T. is supported by a Ruth L. Kirschstein National Research Service Award (F32 HL129989). We thank Dr. Heidi K. Privett for providing HG2 design scripts and computational model, and Dr. Justin T. Biel for assistance with initial processing of X-ray datasets. Data collection at Beamline 8.3.1 at the Advanced Light Sources is supported by the University of California Office of the President, Multicampus Research Programs and Initiatives grant MR-15-328599, the Program Breakthrough Biomedical Research (which is partially funded by the Sandler Foundation), the National Institutes of Health (R01 GM124149 and P30 GM124169), Plexxikon Inc., and the Integrated Diffraction Analysis Technologies program of the US Department of Energy Office of Biological and Environmental Research. The Advanced Light Source (Berkeley, CA) is a national user facility operated by Lawrence Berkeley National Laboratory on behalf of the US Department of Energy under contract number DE-AC02-05CH11231, Office of Basic Energy Sciences. This research was enabled in part by support provided by Compute Ontario (https://computeontario.ca) and Compute Canada (https://www.computecanada.ca).

## Author contributions

A.B. and R.A.C. conceived the project. A.B., N.Z., E.N., N.M., and L.L. purified proteins. A.B. and N.Z. performed enzyme kinetics experiments. R.A.C. and M.C.T. crystallized proteins and performed X-ray diffraction experiments. A.B., R.V.R., M.C.T., and R.A.C. performed refinements. M.C.T. and J.S.F. designed X-ray crystallography experiments. R.V.R. and A.B. performed computational design experiments. R.A.C. wrote the manuscript. A.B., R.V.R., and M.C.T. edited the manuscript.

## Competing interests

The authors declare no competing interests.
