## [Peer Review File · Nature Communications]

REVIEWER COMMENTS

Reviewer #1 (Remarks to the Author):

Overall, I think this is a nice study and an important contribution to the field. The structural biology and computational design are largely flawless and of excellent quality. I disagree a little with some of the interpretation and there are some issues with the kinetics - I have included some suggestions below that might help improve the paper.

First, the title is a little misleading, because the analysis of the evolution didn't really guide the design of an efficient biocatalyst? It was more that the authors identified that the use of ensembles (from the original design) could produce vastly superior designs. So maybe the title could be refined a little.

The first part of the paper is following the evolution of activity with structural analysis. This has already been done to some extent by Hilvert, who identified increased complementarity, improved preorganization of the catalytic Asp and the new catalytic Glu as key contributions to improved catalysis. Some additions to this are included here, such as stabilisation of a cis bond to prevent formation of the less optimal trans configuration, some changes near the substrate entrance and stabilisation of the catalytic Gln. These observations help provide more detail to the work by Hilvert et al, but I don't think they substantially change the original conclusions.

The second half of the paper looks at taking an ensemble of structures from a refinement of HG3 for design, which is very interesting in that it shows various non-ground state (I.e. not the crystal structure) backbone conformations from the ensemble can generate the "optimal" solution. This has important implications for protein design.

Specific comments:

Intro: I think there needs to be a bit more explicit discussion of what has been done previously, in the sense that Hilvert et al have analysed the structure of the evolved variant and identified three important changes: shape complementarity of the pocket, alignment/preorganization of Asp127 and the new catalytic group.

Ln 74: the kemp elimination is a concerted base-abstraction and ring-opening reaction, not just deprotonation.

Ln 85: HG2 sort of comes out of nowhere here - could some additional explanation of what this variant is be provided? It's absent from the introduction, etc. (I worked it out when reading the SI but would be good to introduce earlier just to be clear).

Ln 89: the chemical name of the TSA should be provided here, and used throughout (fine to give in an abbreviation), rather than calling it TSA - there are many possible TSAs for this TS.

Ln 134: I'm not sure if it can be said that the cis bond between 83 and 84, and the hydrogen bond to Gln50 accounts for the majority of the 12-fold catalytic enhancement. The H-bond with the backbone were previously highlighted by Hilvert et al and the entropic benefit to the Gln here in place of His, so really it is the introduction of the new catalytic gln (concomitant with the stabilisation of the cis bond) that is important. But it is essential that this peptide could sample the cis conformation to allow this to be stabilised - which is an important addition that this study makes.

Ln 137: Attributing the change between 3.7 and 3.17 to changes in the conformational ensemble is very difficult because of all of the other mutations. For example, 275/276, which were distal and

included in HG4 are incorporated within this evolutionary span. How much of the 6-fold increase is due to them? From a comparison between HG4 and 3.7 it looks like as much as 4-fold of the 6-fold increase could be due to the effect of these two mutations. This would make the additional stabilisation of Gln50 ~2-fold. This is just to point out that this type of analysis is very difficult without looking closely at the effects of groups of mutations in identical backgrounds. So, in this section, I suggest reducing the attribution of the increased activity on the stabilisation of Gln50 because I don't think it's fully consistent with the results (I think there are probably a number of changes taking place (including stabilisation of Gln50) - 10 generations is a lot - that all have small effects). I think it probably does have an effect, I am a big advocate of the idea, but here it looks like it is probably more of a fine-tuning change rather than the dominant cause of the increase in activity in this instance - I think it is actually nice to see that there are many mechanisms by which enzymes can improve.

Ln 183: Does the previously published high resolution unbound structure of HG3.17 sample similar conformations?

Ln 200: It says in the abstract that you made "HG4, an efficient biocatalyst (k_{cat}/K_M 120,000 $M^{-1}s^{-1}$) containing active-site mutations found during evolution but not distal ones". There's no clear figure where all the mutations are shown on the structure - maybe they could be labelled by number or something in a figure with balls as in Fig 1? Regardless, it clearly does include distal mutations (275/276), so this statement probably needs to be modified.

Ln 210: "distal mutations in HG3.17 contribute little to its catalytic efficiency" The turnover rate of 3.17 is 50000 ($M^{-1}s^{-1}$) higher than HG4 - this difference alone is >300-fold the k_{cat}/K_M of HG3 from Table 1. I also think the activity of 3.17 is underestimated in this work (see below) It can be deceptive referring to activities in terms of fold changes (it is easy to get a 100-fold increase of a very low activity, exceptionally hard for a high activity). It's important to note the magnitude of the rate constants as well as fold changes. So, here I think it is probably wrong to state that the distal mutations contribute little to the catalytic efficiency. They clearly contribute a lot - because it gets increasingly difficult to increase activity as activity levels rise owing to diminishing returns - see Tokuriki et al Nat. Commun. 3:1257 doi: 10.1038/ncomms2246 (2012).

Ln 211: If HG4 is better preorganized, and the remote mutations contribute little, why is it less active? This also seems to contradict the previous section, where the stabilisation of Glu by non-active site mutations was proposed to contribute to the 6-fold increase in activity from 3.7. Just looking at figure 3, there does seem to be differences between 3.17 and hg4 - there is a lot of additional atomic displacement in HG4 in loops near the active site. If the k_{cat}/K_M of 3.17 is closer to the previously published value, this could make more sense - the remote mutations could well have an effect by stabilising these active site loops and possibly contributing something around 2 fold increase in activity.

Ln 215: There is discussion of HG2, which I'm not clear on, but why isn't HG3 the more relevant comparison? Could this section be expanded a little to be more clear?

Ln 235: "generated using molecular dynamics restrained by the HG3 diffraction data (Methods)" Is this just another way of saying crystallographic ensemble refinement? Maybe make this clear. This should be expanded to show more specifically what proportion of states within the ensemble worked, and (if possible) an explanation for why (do they cluster together? Are there specific rearrangements in the main chain that are required?). This aspect of the work is the most novel in my opinion and could be expanded. However, beginning the design from HG3 would have been impossible because this is the design - could the authors explain whether it would be possible to improve the original design process by using an ensemble of conformations generated by the 1GOR template?

Ln 311 and SI: I have some questions regarding the kinetics. k_{cat} and K_m are not reported and I'm unsure how the error was calculated for the k_{cat}/K_m . Looking at the curves in the SI - these reactions are not reaching a plateau, which makes calculation of the Michaelis constant and k_{cat} difficult (the errors would be very large from curve fitting). Are the multiple curves repeats, which are used to calculate the error? If that is the case, I have serious reservations about HG3.17, which is critical for some of the discussion above. You can see from the two graphs that the predicted vastly different k_{cat} and K_m values - the one on the left looks like a turnover number of ~ 150 , K_m of ~ 1.2 mM, whereas the one on the right is almost linear and impossible to predict, other than being much higher. It looks like there is a major error here - you can see that the 0.5 mM point in the left graph is over double the graph on the right. It could be that the k_{cat}/K_m that is calculated from each curve is fairly close, by coincidence, but the individual k_{cat} and K_m values would differ drastically - so just calculating the mean and SD of this really underestimates the error. Similarly, HG3 looks like it is underestimated - the curve of the left is plateauing much faster than the curve on the right, which explains the divergence from the Hilvert data again. This will affect some of the fold comparisons.

The more common way to do this would be to plot all the data on a single graph, fit a curve to the data and use the error of the curve fitting to estimate the error.

However, in the Hilvert paper you can see the K_m values are in the order 2-3 (8 for HG3.7) mM, so the concentration range (< 2 mM) used here is not sufficient to derive the K_m using M-M kinetics (which is why the slopes are mostly linear - it is well below the K_m). However, all is not lost - you can obtain estimates of k_{cat}/K_m using: $k_{cat}/K_m(\text{estimated}) = V_0/([S].[Et])$ where $[Et]$ and $[S]$ are the starting enzyme and substrate concentrations respectively, and V_0 is the initial velocity of the reaction - provided $[S] \ll K_m$. So since in this study you only really want to compare k_{cat}/K_m , this might be ok. It looks pretty close to what you/Hilvert are getting anyway in terms of relative values - at 0.5 mM, HG3 ~ 0.6 ; 3b ~ 1.2 ; 3.7 ~ 15 ; 3.14 ~ 25 ; 3.17 ~ 75 ; HG4 ~ 45 . The alternative would be to repeat the kinetics with concentration range that extends beyond the K_m , i.e. to a region where the activity is clearly plateauing (to get an accurate estimate of the k_{cat} and K_m).

If any of this is unclear, or I have misunderstood anything, I am happy to discuss with authors,

Colin Jackson

Reviewer #2 (Remarks to the Author):

Paper by Broom & Rakotoharisoa et al. describes an interesting and successful approach to overcome known deficiencies in the computational methodologies employed for de novo enzyme design. The authors have demonstrated that combining directed evolution approach with crystallographically-derived ensemble of models provides a possibility to assess the conformational flexibility of the backbone atoms. This approach has proven successful and provides a valuable input for the development of better enzyme design protocols which in my eyes merits publication.

Only a few comments concerning the crystallographic work.

Z-score based analysis of atomic displacement factors raises no questions. However description of data collection strategy is not clear. As radiation damage is a critical issue in case of room temperature crystallography, it should be addressed in more details despite the fact that the authors have limited a total X-ray dose to not exceed 200 kGy per a single data set. However, as multiple data sets covering a 180 deg wedge of reciprocal space have been merged together to obtain a final set of reduced intensities, radiation damage could potentially influence the mobility of atoms in the crystal lattice and hence have an impact (most likely positive) on ensemble of templates.

1. Were these multiple data sets collected from the same part of the crystal (in particular the whole crystal) or was the crystal translated and re-centered in between in order to use the "fresh"/"not radiation damaged" part (beam size was much smaller than the crystal itself)?
2. In case the second scenario has been used, how the assessment of crystal isomorphism has been performed? In case the first scenario has been used, how severe the radiation damage was and how was it estimated?
3. Based on reported multiplicities one can expect that some crystals were less robust against radiation damage than the others. In general, for crystals belonging to P212121 space group one 180 deg wedge should be enough to obtain the complete data set (I assume strategy option has been used to collect diffraction data as soon as possible and to reduce the risk of radiation damage). Which criteria have been used for assessment of how many data sets collected from a single crystal should be merged? Was it done automatically with Xia2 program?
4. Did the authors try to compare crystal structures obtained from the first wedge of data with the structures obtained based on the "merged" final data set? This comparison could shed the light on "positive/negative" effects of radiation damage/dose accumulation. A quick and dirty approach would be to compare atomic B-factors between the final models and atomic models with "re-refined" B-factors against the first data set (combination of TLS and B-factors upon resetting these to a value below Wilson B).
4. The Sup. Table 3 should include also Wilson B as well as averaged B-factor for protein atoms and solvent molecules and ligands if present.

Editing mistake

Line 110: This result is in contrast with the 110 deposited structure of HG3.17-E47N/N300D, which contains two molecules in a unit cell of..

The authors mean two molecules in the asymmetric part of a unit cell. Two molecules in a unit cell would indicate that the asymmetric unit contains only a half of a molecule what is definitely not true.

Reviewer #3 (Remarks to the Author):

Broom et al. report a retrospective analysis of the molecular changes that occurred during the evolution of a computationally designed Kemp eliminase. Using room temperature x-ray data of several evolutionary intermediates, they show that the active site of the enzyme became increasingly preorganized and the catalytic groups rigidified as activity increased. Subsequent reversion of 9 distal mutations that had accumulated in the final variant (out of 17 total) yielded a new enzyme, HG4, that retained much of the activity of the parent. Perhaps the most interesting aspect of this study, though, is the authors' exploration of whether the same design protocols that were used to create the original HG3 design could recapitulate the HG4 structure. Although this proved possible when the crystal structure of HG4 was used as a template, the original xylanase scaffold proved unsuitable because of differences in backbone conformation. However, by incorporating backbone modeling of the x-ray data into design procedure, the 'correct' transition state binding mode could be reproduced. This is an important result with potentially significant implications for the design field.

Publication of this well written paper in Nature Communications can be recommended pending consideration of the points raised below.

1) The kinetic parameters reported in Table 1 are systematically lower than the previously published values. Why? How was enzyme concentration determined? Was the substrate stock contaminated by product (a potential competitive inhibitor)? Was the pH measured before or after addition of 10% methanol? Since only k_{cat}/K_m values are reported, it is also odd that the authors fit their data using the Michaelis-Menten equation. They should be able to measure very accurate k_{cat}/K_m values by fitting a line through the data obtained at substrate concentrations well below K_m .

2) The peptide bond between residues 83 and 84 is described as a mixture of cis and trans configurations for HG3 and HG3.3b but exclusively cis for the more evolved variants (p. 7). While the electron density shown in Fig. 2 for HG3.3b is pretty convincing, the situation is less obvious for HG3. Since the resolution of the latter structure is the lowest of the lot, is modeling the minor trans configuration really justified?

3) In the discussion of active site preorganization, the authors note that Trp44 and Met237 adopt conformations in the apo protein that would prevent productive binding of the TSA. Is the energetic cost of moving these residues out of the way really expected to be high? Along the same lines, Fig 2e seems to suggest that Trp44 is just as disordered in HG3.17 as in its predecessors (p. 9). Have I missed something?

4) The inclusion of two distal mutations in HG4, namely R275A and W276F, weakens the broad claim that distal mutations in HG3.17 contribute little to catalytic efficiency (p. 10). Is the variant lacking these two mutations equally active?

5) For optimization of the HG4 sequence using MD-generated backbone ensembles, how important were the restraints imposed by the HG3 diffraction data? Were controls run simply using MD without the restraints? How did the results compare? Although diffraction data for this system were available, thanks to the authors, that won't generally be the case. Can unrestrained MD simulations be used directly to generate ensembles of relevant backbone templates to improve the design predictions? Some discussion of this issue is important.

6) Producing more active computational enzymes is an important goal in the design community which could minimize or eliminate the need for tedious optimization by directed evolution. Although the authors have shown that they can computationally design a de novo Kemp eliminase with native-like catalytic efficiencies when they already know what an active enzyme looks like, it is not yet clear to what extent their findings will translate to other scaffolds and other reactions if this information is not available ahead of time.

Response to Reviewers and List of Changes

Reviewer #1 (Remarks to the Author):

Overall, I think this is a nice study and an important contribution to the field. The structural biology and computational design are largely flawless and of excellent quality. I disagree a little with some of the interpretation and there are some issues with the kinetics - I have included some suggestions below that might help improve the paper.

Response: We thank Reviewer #1 for the positive comments, and hope to have addressed all concerns with our responses below.

First, the title is a little misleading, because the analysis of the evolution didn't really guide the design of an efficient biocatalyst? It was more that the authors identified that the use of ensembles (from the original design) could produce vastly superior designs. So maybe the title could be refined a little.

Response: We have changed the title to better reflect the findings of our work:

Ensemble-based enzyme design can recapitulate the effects of laboratory directed evolution in silico

The first part of the paper is following the evolution of activity with structural analysis. This has already been done to some extent by Hilvert, who identified increased complementarity, improved preorganization of the catalytic Asp and the new catalytic Glu as key contributions to improved catalysis. Some additions to this are included here, such as stabilisation of a cis bond to prevent formation of the less optimal trans configuration, some changes near the substrate entrance and stabilisation of the catalytic Gln. These observations help provide more detail to the work by Hilvert et al, but I don't think they substantially change the original conclusions.

The second half of the paper looks at taking an ensemble of structures from a refinement of HG3 for design, which is very interesting in that it shows various non-ground state (i.e. not the crystal structure) backbone conformations from the ensemble can generate the "optimal" solution. This has important implications for protein design.

Specific comments:

Intro: I think there needs to be a bit more explicit discussion of what has been done previously, in the sense that Hilvert et al have analysed the structure of the evolved variant and identified three important changes: shape complementary of the pocket, alignment/preorganization of Asp127 and the new catalytic group.

Response: Although we have not explicitly named Hilvert and coworkers in our introduction, his important contributions are already cited and described in this section (reference #15):

"During evolution, active-site residues, including designed catalytic amino acids, were often mutated, leading to enhanced catalysis via optimization of catalytic contacts, ligand binding modes, and transition-state complementarity.^{13-15"}

Nevertheless, we have reworded the statement above to better match the reviewer's suggested wording:

"During evolution, active-site residues, including designed catalytic amino acids, were often mutated, leading to enhanced catalysis via introduction of new catalytic groups, optimization of catalytic contacts and ligand binding modes, and enhanced transition-state complementarity of the binding pocket.^{13-15"}

We have also dedicated a complete subsection of the Results ("*HG series of Kemp eliminases*"), along with accompanying figure (Figure 1) to explain in detail the findings of Hilvert and coworkers. We felt it more appropriate to include this essential information at the beginning of the Results section instead of the introduction for the following reasons:

1. It describes many important technical details that we feel are more appropriate to include in the section where we provide the rationale for our work.
2. We did not want our introduction to focus on the Hilvert study (Blomberg *Nature* 2013) because studies from other groups have also reported similar findings (e.g., Khersonsky *PNAS* 2012 and Khersonsky *J. Mol. Biol.* 2011).
3. It would make the introduction unnecessarily long, and break the flow of our narrative.

For these reasons, we prefer to maintain the current organization of our manuscript.

Ln 74: the kemp elimination is a concerted base-abstraction and ring-opening reaction, not just deprotonation.

Response: We have modified the description of the Kemp elimination reaction to more accurately reflect its mechanism:

"This artificial enzyme catalyzes the concerted deprotonation and ring-opening of 5-nitrobenzoxazole into the corresponding *o*-cyanophenolate".

We prefer to use "deprotonation" instead of "abstraction" to avoid possible confusion with certain abstraction reactions that involve the removal of a hydrogen atom by a radical.

Ln 85: HG2 sort of comes out of nowhere here - could some additional explanation of what this variant is be provided? It's absent from the introduction, etc. (I worked it out when reading the SI but would be good to introduce earlier just to be clear).

Response: We agree that confusion exists between HG2 and HG3 because HG2 is the original *in silico* design that resulted from the enzyme design calculation performed by Privett et al. (reference #5), while HG3 is a higher-activity mutant of HG2 (S265T) that was engineered post-design to reduce the active-site conformational heterogeneity observed by molecular dynamics analysis of HG2. HG3 was then used by Hilvert and coworkers as the starting template for directed evolution. While the crystal structure of HG2 was solved by Privett et al., the structure of HG3 had not been reported until now. To clarify, we have introduced HG2 earlier in the Results section, and referenced the supplementary figure that describes its relation to HG3 (old Supplementary Figure 2, now renamed Supplementary Figure 1):

"HG3.17 was evolved from HG3, a higher-activity mutant (S265T) of the *in silico* design HG2 (Supplementary Figure 1) that was engineered post-design to reduce the active-site conformational heterogeneity observed by molecular dynamics analysis of HG2."

We have also modified the legend to Supplementary Figure 1 to clarify:

Supplementary Figure 1. HG2 is the direct precursor to HG3. For the crystal structures of HG2 and HG3, the 2Fo-Fc map is shown in volume representation at two contour levels: $0.5 \text{ e}\text{\AA}^{-3}$ and $1.5 \text{ e}\text{\AA}^{-3}$ in light and dark blue, respectively. HG2 was designed *in silico* to stabilize the transition state (TS) via catalytic contacts (dashed lines) with the D127 base and the S265 hydrogen bond donor (Left panel). However, its crystal structure (PDB ID: 3NYD) showed that the transition state analogue 6-nitrobenzotriazole (6NT) was bound in two alternate orientations (Middle panel). In the catalytically productive pose, the acidic N–H bond of 6NT that mimics the cleavable C–H bond of the substrate is located within H-bonding distance to the carboxylate oxygen of D127, and the nitro group is close to S265 (found to adopt two conformations in the crystal structure). In the catalytically non-productive pose, 6NT is flipped, which positions its nitro group closer to K50, and its acidic N–H bond far from the side chain of D127. **To increase activity, Privett et al. introduced the S265T mutation into HG2 post-design to reduce the active-site conformational heterogeneity that was observed by molecular dynamics analysis, leading to the higher-activity variant HG3 (Right panel).** Of note, no density for the non-productive binding pose of 6NT was observed in the HG3 crystal structure reported here.

Ln 89: the chemical name of the TSA should be provided here, and used throughout (fine to give in an abbreviation), rather than calling it TSA - there are many possible TSAs for this TS.

Response: We have replaced all occurrences of TSA with 6NT, which is the PDB ID for the 6-nitrobenzotriazole transition-state analogue used here.

Ln 134: I'm not sure if it can be said that the *cis* bond between 83 and 84, and the hydrogen bond to Gln50 accounts for the majority of the 12-fold catalytic enhancement. The H-bond with the backbone were previously highlighted by Hilvert et al and the entropic benefit to the Gln here in place of His, so really it is the introduction of the new catalytic gln (concomitant with the stabilisation of the *cis* bond) that is important. But it is essential that this peptide could sample the *cis* conformation to allow this to be stabilised - which is an important addition that this study makes.

Response: We apologize for the confusion. What we meant to say is not that the *cis* peptide bond between 83 and 84 accounts for the majority of the 12-fold catalytic enhancement but that it is the introduction of the Gln50 residue stabilized in a catalytically-productive conformation via a hydrogen bond to this *cis* peptide that accounts for the majority of the rate enhancement. This assumption is supported by a recent study from Hilvert and coworkers (Kries *ACS Catalysis* 2020), who showed that the HG3.17-Q50H single mutant displays a 16-fold decrease in catalytic efficiency relative to HG3.17.

To clarify, we have reworded the following statement:

“This hydrogen bonding interaction helps to lock Gln50 in a conformation that is properly oriented to stabilize negative charge buildup on the phenolic oxygen at the transition state, likely accounting for the majority of the 12-fold catalytic efficiency enhancement observed in HG3.7 relative to HG3.3b.”

to:

“This hydrogen bonding interaction helps to lock Gln50 in a conformation that is properly oriented to stabilize negative charge buildup on the phenolic oxygen at the transition state. Introduction of this new catalytic group in a catalytically-productive conformation likely accounts for the majority of the 12-fold enhancement in k_{cat}/K_M observed in HG3.7 relative to HG3.3b, a hypothesis that is supported by the 16-fold decrease in k_{cat}/K_M observed when the Q50H mutation is introduced into HG3.17 [Kries ACS Catalysis 2020].”

Ln 137: Attributing the change between 3.7 and 3.17 to changes in the conformational ensemble is very difficult because of all of the other mutations. For example, 275/276, which were distal and included in HG4 are incorporated within this evolutionary span. How much of the 6-fold increase is a due to them? From a comparison between HG4 and 3.7 it looks like as much as 4-fold of the 6-fold increase could be due to the effect of these two mutations. This would make the additional stabilisation of Gln50 ~2-fold. This is just to point out that this type of analysis is very difficult without looking closely at the effects of groups of mutations in identical backgrounds. So, in this section, I suggest reducing the attribution of the increased activity on the stabilisation of Gln50 because I don't think it's fully consistent with the results (I think there are probably a number of changes taking place (including stabilisation of Gln50) - 10 generations is a lot - that all have small effects). I think it probably does have an effect, I am a big advocate of the idea, but here it looks like it is probably more of a fine-tuning change rather than the dominant cause of the increase in activity in this instance - I think it is actually nice to see that there are many mechanisms by which enzymes can improve.

Response: Reviewer 1 is correct to note that the change to the conformational ensemble and resulting Q50 rigidification that we observed during evolution from HG3.7 to HG3.17 is not the sole contributor to the increased activity. Other effects such as widening of the active-site entrance by mutation to 275/276 as well as improvement of active-site pre-organization also contribute greatly. We apologize if we gave the impression that we attributed the 6-fold increase of activity between HG3.7 and HG3.17 solely to rigidification of Q50.

To reduce attribution of the increased activity on stabilisation of Gln50, we have added the following sentence between the paragraph where we describe Q50 rigidification and the one where we describe changes in active-site pre-organization:

“Although increasing rigidity of the Gln50 catalytic residue from HG3.7 to HG3.17 likely contributes to enhanced catalysis, other structural effects were investigated.”

Ln 183: Does the previously published high resolution unbound structure of HG3.17 sample similar conformations?

Response: We are unaware of such a structure, which we could not find on the Protein Data Bank (all deposited structures of HG3.17 have a bound transition-state analogue). Our current manuscript is, to the best of our knowledge, the first report of the unbound structure of HG3.17.

Ln 200: It says in the abstract that you made “HG4, an efficient biocatalyst (k_{cat}/K_M 120,000 M⁻¹s⁻¹) containing active-site mutations found during evolution but not distal ones”. There's no clear figure where all the mutations are shown on the structure - maybe they could be labelled by number or something in a figure with balls as in Fig 1? Regardless, it clearly does include distal mutations (275/276), so this statement probably needs to be modified.

Response: We have added a panel to Figure 1 (panel e) showing the mutation sites on HG4. We have also modified the abstract to clarify what we mean by “active-site mutations found during evolution but not distal ones”:

“Based on these observations, we engineered HG4, an efficient biocatalyst (k_{cat}/K_M 103,000 $M^{-1} s^{-1}$) containing key first and second-shell mutations found during evolution.”

Ln 210: “distal mutations in HG3.17 contribute little to its catalytic efficiency” The turnover rate of 3.17 is 50000 ($M^{-1} s^{-1}$) higher than HG4 - this difference alone is >300-fold the k_{cat}/K_M of HG3 from Table 1. I also think the activity of 3.17 is underestimated in this work (see below) It can be deceptive referring to activities in terms of fold changes (it is easy to get a 100-fold increase of a very low activity, exceptionally hard for a high activity). It’s important to note the magnitude of the rate constants as well as fold changes. So, here I think it is probably wrong to state that the distal mutations contribute little to the catalytic efficiency. They clearly contribute a lot - because it gets increasingly difficult to increase activity as activity levels rise owing to diminishing returns - see Tokuriki et al Nat. Commun. 3:1257 doi: 10.1038/ncomms2246 (2012).

Response: This is a fair point. Accordingly, we have replaced the following statement:

“Interestingly, both HG4 and HG3.17 have catalytic efficiencies on the order of $10^5 M^{-1} s^{-1}$ despite the fact that the former enzyme contains less than half of the latter’s mutations, demonstrating that distal mutations in HG3.17 contribute little to its catalytic efficiency.”

With:

“However, HG4 is approximately 20% less active than HG3.17, demonstrating that the additional 9 mutations found in the latter enzyme, most of which are distal to the active site, play a role in enhancing catalytic efficiency.”

Ln 211: If HG4 is better preorganized, and the remote mutations contribute little, why is it less active? This also seems to contradict the previous section, where the stabilisation of Glu by non-active site mutations was proposed to contribute to the 6-fold increase in activity from 3.7. Just looking at figure 3, there does seem to be differences between 3.17 and hg4 - there is a lot of additional atomic displacement in HG4 in loops near the active site. If the k_{cat}/K_M of 3.17 is closer to the previously published value, this could make more sense - the remote mutations could well have an effect by stabilising these active site loops and possibly contributing something around 2 fold increase in activity.

Response: We agree and have replaced the statement about distal mutations in HG3.17 contributing little to its catalytic efficiency with one emphasizing that although HG4 is better pre-organized than HG3.17, it is still less active (see above).

Regarding the difference between our measured k_{cat}/K_M value for HG3.17 ($126,000 M^{-1} s^{-1}$) to the previously published value ($230,000 M^{-1} s^{-1}$), it is unclear what is the cause of this discrepancy (see our response to Reviewer #3 Comment #1 for discussion of this topic). We typically measure our own kinetic parameters in order to make comparisons between mutants instead of simply referring to published literature data. This is done to ensure that all assays and enzyme samples are prepared and analyzed in the same way, allowing for an internally consistent comparison of enzyme variants. Although the absolute values of our reported catalytic efficiencies are lower than those reported by Hilvert and coworkers, their relative values are similar. To remain internally consistent, we prefer to use our measured k_{cat}/K_M value

for HG3.17 (and for all other HG variants) instead of the value reported by Hilvert and coworkers when drawing conclusions on our newly-engineered variant HG4.

Regarding the proposed effect on loop flexibility of the distal mutations found in HG3.17 but not HG4, we would like to note the difference in crystal form between these two proteins, which makes it impossible to comment on this with certainty, since the crystal packing environment in HG3.17 appears more restrictive of this region than for HG4. For this reason, we limited our analysis to the 87–90 loop, which becomes more rigid as activity increases in the enzymes that adopt a similar crystal form (HG3 -> HG3.3b -> HG3.7 -> HG3.14 -> HG4).

Ln 215: There is discussion of HG2, which I'm not clear on, but why isn't HG3 the more relevant comparison? Could this section be expanded a little to be more clear?

Response: As explained above, the confusion between HG2 and HG3 arises from the fact that HG2 and not HG3 is the result of the enzyme design calculation (HG3 was engineered post-design following molecular dynamics analysis of HG2). Therefore, it would be factually incorrect to state that we used a design protocol similar to the one that produced HG3 instead of the current statement (*"Given that all but one mutation (G82A) in HG4 are found at sites that were optimized during design of HG2, we investigated whether the HG4 structure could be accurately predicted using a computational protocol similar to the one that produced HG2"*). We hope that the statement we added earlier to explain the lineage of HG3 helps to address any ambiguity about why we discuss HG2 instead of HG3 in this paragraph.

Ln 235: "generated using molecular dynamics restrained by the HG3 diffraction data (Methods)" Is this just another way of saying crystallographic ensemble refinement? Maybe make this clear.

Response: To clarify, we have changed the statement to:

"we generated backbone ensembles using molecular dynamics restrained by the HG3 or 1GOR diffraction data, also known as ensemble refinement (Methods)"

This should be expanded to show more specifically what proportion of states within the ensemble worked, and (if possible) an explanation for why (do they cluster together? Are there specific rearrangements in the main chain that are required?). This aspect of the work is the most novel in my opinion and could be expanded. However, beginning the design from HG3 would have been impossible because this is the design - could the authors explain whether it would be possible to improve the original design process by using an ensemble of conformations generated by the 1GOR template?

Response: To answer these questions, we generated an ensemble of conformations using ensemble refinement on 1GOR, and used the resulting templates to perform additional design calculations. The best model obtained using the 1GOR ensemble differs significantly from the HG4 crystal structure and is destabilized by approximately 35 kcal/mol (Figure 4g). However, it is ~10 kcal/mol more stable than the model obtained on the 1GOR crystal structure (Figure 4b). These results show that the use of an ensemble of conformations generated by ensemble refinement of the 1GOR template does not substantially improve the design process. We have expanded the design aspect of our work by (i) writing a new paragraph in the *"Computational design of HG4"* subsection of the "Results", (ii) modifying Supplementary Figure 5 (now renamed Supplementary Figure 6) to show more specifically what proportion of states within the ensemble worked, and (iii) creating a new figure (Supplementary Figure 7) to provide an explanation for why (see below):

“To address issues arising from the use of a single fixed backbone template, we generated backbone ensembles using molecular dynamics restrained by the HG3 or 1GOR diffraction data, also known as ensemble refinement (Methods), and used the resulting templates to optimize rotamers for the HG4 sequence. We were able to recapitulate the correct transition-state binding mode on several individual ensemble members derived from the HG3 crystallographic data, with energies comparable to that of the HG4 crystal structure (Figure 4e–f, Supplementary Figure 6). However, use of an ensemble derived from the 1GOR diffraction data did not allow recapitulation of the crystallographic transition-state binding mode (Figure 4g) although it did yield several computational models displaying improved energy (Supplementary Figure 6). The inferior performance of the 1GOR-derived ensemble compared to the HG3 ensembles likely results from differences in conformational heterogeneity within the ensemble, specifically at position 83 (Supplementary Figure 7).”

and

“The better predictive ability of the HG3-derived ensembles prepared using crystallographic restraints likely results from their lower deviation from the HG4 crystal structure (≈ 0.4 Å, Supplementary Figure 7), which we previously showed to be necessary for an ensemble to represent a physically valid model of the target protein fold.²⁴ Overall, these results suggest that computational enzyme design with a crystallographically-derived backbone ensemble derived from a low activity enzyme could obviate the need for directed evolution by allowing catalytically-competent sub-states to be sampled during the design procedure.”

Supplementary Figure 6. Energy of HG4 design models generated on various backbone templates. Rotamers for the HG4 sequence and its associated transition state binding pose were optimized (Methods) on individual backbone templates (bars). These ensembles of backbone templates were generated using molecular dynamics (MD) constrained or not by the diffraction data. In all cases, n indicates the total number of templates in the ensemble. Green bars indicate templates that yielded design models with transition-state binding poses within 0.7 Å root-mean-square deviation from the crystallographic binding pose. Blue and red bars indicate design models obtained from the HG4 with bound 6NT (-186.4 kcal/mol) or 1GOR (-141.6 kcal/mol) crystal structures, respectively. All HG3-derived ensembles yielded at least one HG4 design model with an accurate transition-state binding pose (i.e., root-mean-square deviation < 0.7 Å) and a more favorable energy than that obtained on the 1GOR template. However, the ensemble prepared from the HG3 (+) 6NT data outperforms all others: it yielded the lowest energy model and the highest number of models with accurate transition-state binding poses (21 out of 84).

Supplementary Figure 7. Backbone ensembles. The crystal structure of HG4 (+) 6NT (magenta) is superimposed with members of ensembles (grey) generated by ensemble refinement or unconstrained molecular dynamics (MD) starting from the HG3 (+) 6NT, HG3 (-) 6NT, or 1GOR crystal structures. Spheres indicate alpha carbons of key active-site residues. Ensemble properties, such as the average root-mean-square backbone coordinate deviation between pairs of ensemble members (diversity) or average root-mean-square backbone coordinate deviation from the HG4 (+) 6NT crystal structure (deviation) are indicated. In all cases, n indicates the number of templates per ensemble. Several templates from HG3-derived ensembles enable recapitulation of the crystallographic transition-state binding pose (see Figure 4 and Supplementary Figure 6) due to the ability of the residue 83 backbone to sample conformations that eliminate steric clashes with this ligand, and that are similar to the one observed in the HG4 (+) 6NT crystal structure (i.e., there is overlap of magenta and grey spheres at position 83). This is not the case for the 1GOR-derived ensemble, in which the position of the residue 83 backbone does not shift substantially, preventing it from adopting a conformation necessary to allow the crystallographic transition-state binding pose found in the HG4 (+) 6NT crystal structure (i.e., there is no overlap of magenta and grey spheres at position 83). Of note, the ensemble generated by unconstrained MD (bottom left) samples a range of conformations with higher

diversity and deviation than the corresponding one generated from ensemble refinement (top right), and with evenly-distributed conformational heterogeneity throughout the structure. By contrast, conformational heterogeneity in ensembles generated using crystallographic restraints is unevenly distributed, being higher in specific structural elements.

Ln 311 and SI: I have some questions regarding the kinetics. k_{cat} and K_m are not reported and I'm unsure how the error was calculated for the k_{cat}/K_m . Looking at the curves in the SI - these reactions are not reaching a plateau, which makes calculation of the Michaelis constant and k_{cat} difficult (the errors would be very large from curve fitting). Are the multiple curves repeats, which are used to calculate the error? If that is the case, I have serious reservations about HG3.17, which is critical for some of the discussion above. You can see from the two graphs that they predict vastly different k_{cat} and K_m values - the one on the left looks like a turnover number of ~ 150 , K_m of ~ 1.2 mM, whereas the one on the right is almost linear and impossible to predict, other than being much higher. It looks like there is a major error here - you can see that the 0.5 mM point in the left graph is over double the graph on the right. It could be that the k_{cat}/K_m that is calculated from each curve is fairly close, by coincidence, but the individual k_{cat} and K_m values would differ drastically - so just calculating the mean and SD of this really underestimates the error. Similarly, HG3 looks like it is underestimated - the curve of the left is plateauing much faster than the curve on the right, which explains the divergence from the Hilvert data again. This will affect some of the fold comparisons.

Response: Thank you for bringing this issue to our attention. We have reanalyzed all of our raw kinetic data and realized that a copy-paste error was made when we copied HG3.17 rates from our Excel spreadsheet to the GraphPad software used to generate the Michaelis-Menten plots (data for all other variants was correct). We have now fixed this issue and report new Michaelis-Menten plots (see below) combining all replicates measured from multiple independent enzyme batches on a single graph per variant, as suggested by Reviewer #1 in the next comment. We apologize for this mistake.

Supplementary Figure 2. Steady-state kinetics. Michaelis–Menten plots of normalized initial rates as a function of 5-nitrobenzisoxazole (5NBZ) concentrations are shown. Data represent the average of six or nine individual replicate measurements from two or three independent protein batches, with error bars indicating the SEM. Saturation was not achieved for any enzyme at the substrate’s solubility limit (2 mM). Therefore, only k_{cat}/K_M values are reported on Table 1, and these values were calculated using linear regression of rates measured at the three or four lowest substrate concentrations.

As explained in the figure legend above, we did not use the Michaelis-Menten equation to calculate individual k_{cat} or K_M values because saturation was not achieved for any Kemp eliminase. This is due to the solubility limit of the 5-nitrobenzisoxazole substrate in 10% methanol, which was determined by Hilvert and coworkers to be 2.2 mM (Blomberg *Nature* 2013) – our experimental observations support this number. Reviewer #1 is correct that using the Michaelis-Menten equation without achieving saturation would result in very large errors in k_{cat} , K_M , and k_{cat}/K_M . Therefore, we used linear regression to calculate k_{cat}/K_M from rates obtained at low substrate concentrations, as described below.

The more common way to do this would be to plot all the data on a single graph, fit a curve to the data and use the error of the curve fitting to estimate the error.

Response: We have combined all data from each variant into a single graph per enzyme (see Supplementary Figure 2 above), but only used the rates obtained at the 3 or 4 lowest substrate concentrations ($[S] \ll K_M$) to calculate k_{cat}/K_M by linear regression (as suggested by Reviewer #3, comment #1). To estimate the error on these k_{cat}/K_M values, we used the error of the curve fitting as suggested by Reviewer #1 (see revised Table #1 below). Importantly, these new numbers are within error of the numbers in our previous draft, with the exception of that for HG3.17, which has decreased from $170000 \pm 20000 \text{ M}^{-1}\text{s}^{-1}$ to $126000 \pm 9000 \text{ M}^{-1}\text{s}^{-1}$. This difference is due to the copy-paste error that we described above.

Table 1. Kinetic parameters of Kemp eliminases

Enzyme	Mutations from HG3 ^a	k_{cat}/K_M ($\text{M}^{-1} \text{s}^{-1}$) ^b
HG3	-	146 ± 6 (1300)
HG3.3b	V6I K50H M84C S89R Q90D A125N	2200 ± 100 (5400)
HG3.7	V6I Q37K K50Q M84C S89R Q90H A125N	27000 ± 2000 (37000)
HG3.14	V6I Q37K K50Q G82A M84C Q90H T105I A125T T142N T208M T279S D300N	52000 ± 1000 (70000)
HG3.17	V6I Q37K N47E K50Q G82A M84C S89N Q90F T105I A125T T142N T208M F267M W275A R276F T279S D300N	126000 ± 9000 (230000)
HG4	K50Q G82A M84C Q90F A125T F267M W275A R276F	103000 ± 4000

^a Mutations in italics occurred at sites optimized during computational design of HG2.⁵

^b Individual parameters K_M and k_{cat} could not be determined accurately because saturation was not possible at the maximum substrate concentration tested (2 mM), which is the substrate's solubility limit (Supplementary Fig. 2). Catalytic efficiencies (k_{cat}/K_M) were obtained by linear regression of six or nine individual replicates of rates measured at the three or four lowest substrate concentrations from two or three independent protein batches. Errors of curve fitting are provided. Values in parentheses are from Blomberg *et al.*¹⁵

However, in the Hilvert paper you can see the K_M values are in the order 2-3 (8 for HG3.7) mM, so the concentration range (<2 mM) used here is not sufficient to derive the K_M using M-M kinetics (which is why the slopes are mostly linear - it is well below the K_M). However, all is not lost - you can obtain estimates of k_{cat}/K_M using: $k_{\text{cat}}/K_M(\text{estimated}) = V_0/([S] \cdot [Et])$ where $[Et]$ and $[S]$ are the starting enzyme and substrate concentrations respectively, and V_0 is the initial velocity of the reaction - provided $[S] \ll K_M$. So since in this study you only really want to compare k_{cat}/K_M , this might be ok. It looks pretty close to what you/Hilvert are getting anyway in terms of relative values - at 0.5 mM, HG3 ~0.6; 3b ~ 1.2; 3.7 ~15; 3.14 ~25; 3.17~75; HG4 ~45. The alternative would be to repeat the kinetics with concentration range that extends beyond the K_M , i.e. to a region where the activity is clearly plateauing (to get an accurate estimate of the k_{cat} and K_M).

Response: Reviewer #1 is correct that although the absolute values of our measured catalytic efficiencies are lower than those reported by Hilvert and coworkers, their relative values are similar. As mentioned above, we cannot solubilize the 5-nitrobenzisoazole substrate at concentrations > 2 mM in an aqueous solution containing 10% methanol, preventing us from achieving saturation.

If any of this is unclear, or I have misunderstood anything, I am happy to discuss with authors,

Colin Jackson

Response: Thank you for the rigorous analysis of our work!

Reviewer #2 (Remarks to the Author):

Paper by Broom & Rakotoharisoa *et al.* describes an interesting and successful approach to overcome known deficiencies in the computational methodologies employed for *de novo* enzyme design. The authors have demonstrated that combining directed evolution approach with crystallographically-derived ensemble of models provides a possibility to assess the

conformational flexibility of the backbone atoms. This approach has proven successful and provides a valuable input for the development of better enzyme design protocols which in my eyes merits publication.

Response: We thank Reviewer #2 for the positive comments.

Only a few comments concerning the crystallographic work.

Z-score based analysis of atomic displacement factors raises no questions. However description of data collection strategy is not clear. As radiation damage is a critical issue in case of room temperature crystallography, it should be addressed in more details despite the fact that the authors have limited a total X-ray dose to not exceed 200 kGy per a single data set. However, as multiple data sets covering a 180 deg wedge of reciprocal space have been merged together to obtain a final set of reduced intensities, radiation damage could potentially influence the mobility of atoms in the crystal lattice and hence have an impact (most likely positive) on ensemble of templates.

Response: We agree with Reviewer #2 that managing radiation damage is a key issue in non-cryogenic X-ray crystallography. The reviewer expresses concern over radiation damage due to our strategy of merging multiple data sets, however, contrary to their interpretation this strategy is essential for keeping the effective radiation dose below the critical threshold of 200 kGy provided in the manuscript. We hope that our responses to the referee's specific concerns (below) clarify the nuances surrounding this important concept.

1. Were these multiple data sets collected from the same part of the crystal (in particular the whole crystal) or was the crystal translated and re-centered in between in order to use the "fresh"/"not radiation damaged" part (beam size was much smaller than the crystal itself)?

Response: The unit of Gy is a measure of absorbed energy per unit of mass. Because the density of a protein crystal is essentially unchanged during the data collection, this can also be thought of as a measure of energy absorbed per unit of crystal volume. Therefore, the statement that 200 kGy was used for each dataset necessarily implies that each data set was collected from a different crystal volume. In some cases, if crystals were large enough, multiple data sets were collected from "fresh" regions of the same crystal, while in other cases data sets were collected from different crystals. We have now changed the last sentence of the first paragraph in the "X-ray data collection and processing" subsection of the Methods section to clarify this point, so it now reads:

"Multiple data sets were collected for each enzyme variant, either from different crystals, or if their size permitted, from unique regions of larger crystals."

2. In case the second scenario has been used, how the assessment of crystal isomorphism has been performed? In case the first scenario has been used, how severe the radiation damage was and how was it estimated?

Response: The feasibility of merging data sets (i.e. the isomorphism) was performed empirically, by testing the effects of adding or removing individual data sets on the overall CC1/2 and I/sigma in the high-resolution bins of the merged data set. We have now specified this in the text. We note that since many effects can contribute to non-isomorphism (see Crick and Magdoff, 1956), not all of which involve substantial changes to the lattice parameters, there

is no single reliable metric for assessing “isomorphism,” and an empirical strategy based on the quality of the merged data is generally the most robust. We now mention this process in the manuscript. We have added the following sentence to the second paragraph in the “X-ray data collection and processing” subsection of the Methods section:

“We determined which individual data sets should be combined by evaluating the overall effects of adding or removing individual data sets on the CC1/2 and I/sigma in the high-resolution bins of the merged data set.”

Regarding radiation damage, we take an approach of using a low X-ray dose for each data set (below the typical damage threshold), and sampling 180 degrees of reciprocal space. Spreading the diffracted photons across 180 degrees of reciprocal space circumvents the need to use a data collection “strategy” and the signal-to-noise needed to recover high resolution data is recovered from merging highly redundant data resulting from the collection of multiple data sets.

3. Based on reported multiplicities one can expect that some crystals were less robust against radiation damage than the others. In general, for crystals belonging to P212121 space group one 180 deg wedge should be enough to obtain the complete data set (I assume strategy option has been used to collect diffraction data as soon as possible and to reduce the risk of radiation damage). Which criteria have been used for assessment of how many data sets collected from a single crystal should be merged? Was it done automatically with Xia2 program?

Response: The reviewer is correct about the differences in multiplicity; however, they make an erroneous assumption that it is due to radiation damage. The observed multiplicities for merged data sets are highly dependent on the number of individual data sets that were merged. In our case, for some crystals that diffracted more strongly, fewer data sets were required to converge on a high-resolution data set. Also, for some crystals that were more difficult to obtain, the number of crystal specimens available to measure may have been limiting.

4. Did the authors try to compare crystal structures obtained from the first wedge of data with the structures obtained based on the “merged” final data set? This comparison could shed the light on “positive/negative” effects of radiation damage/dose accumulation. A quick and dirty approach would be to compare atomic B-factors between the final models and atomic models with “re-refined” B-factors against the first data set (combination of TLS and B-factors upon resetting these to a value below Wilson B).

Response: As we describe above, each individual data set was collected with approximately the same X-ray dose, all below the 200 kGy limit. Therefore, the effective dose of the merged data is also 200 kGy or less, as the effective dose for a merged dataset is the weighted average of the doses experienced by the individual data sets (not the sum). Therefore, the analysis the reviewer suggests would not be expected to be informative.

4. The Sup. Table 3 should include also Wilson B as well as averaged B-factor for protein atoms and solvent molecules and ligands if present.

Response: We agree that this should be included, and we have added the requested values.

Editing mistake

Line 110: This result is in contrast with the 110 deposited structure of HG3.17-E47N/N300D, which contains two molecules in a unit cell of..

The authors mean two molecules in the asymmetric part of a unit cell. Two molecules in a unit cell would indicate that the asymmetric unit contains only a half of a molecule what is definitely not true.

Response: We thank the reviewer for catching this mistake. We have changed the following text:

“which contains two molecules in a unit cell of identical space group and similar dimensions”

So that it now reads:

“which contains two molecules in **the asymmetric unit**, with identical space group and similar **unit cell** dimensions”

Reviewer #3 (Remarks to the Author):

Broom et al. report a retrospective analysis of the molecular changes that occurred during the evolution of a computationally designed Kemp eliminase. Using room temperature x-ray data of several evolutionary intermediates, they show that the active site of the enzyme became increasingly preorganized and the catalytic groups rigidified as activity increased. Subsequent reversion of 9 distal mutations that had accumulated in the final variant (out of 17 total) yielded a new enzyme, HG4, that retained much of the activity of the parent. Perhaps the most interesting aspect of this study, though, is the authors' exploration of whether the same design protocols that were used to create the original HG3 design could recapitulate the HG4 structure. Although this proved possible when the crystal structure of HG4 was used as a template, the original xylanase scaffold proved unsuitable because of differences in backbone conformation. However, by incorporating backbone modeling of the x-ray data into design procedure, the 'correct' transition state binding mode could be reproduced. This is an important result with potentially significant implications for the design field.

Publication of this well written paper in Nature Communications can be recommended pending consideration of the points raised below.

Response: We thank Reviewer #3 for the positive comments.

1) The kinetic parameters reported in Table 1 are systematically lower than the previously published values. Why? How was enzyme concentration determined? Was the substrate stock contaminated by product (a potential competitive inhibitor)? Was the pH measured before or after addition of 10% methanol? Since only k_{cat}/K_m values are reported, it is also odd that the authors fit their data using the Michaelis-Menten equation. They should be able to measure very accurate k_{cat}/K_m values by fitting a line through the data obtained at substrate concentrations well below K_m .

Response: Differences in kinetic parameters between studies are not unexpected given that there are many small experimental variations that can add up to create statistically significant

differences (e.g., substrate and/or enzyme concentration and purity, handling of enzyme sample, delays between mixing of reagents and acquisition of initial rates, variations in buffer composition and temperature, etc.). In this study, we took great care to minimize sources of error by using identical protocols to express, purify, quantify, and assay each enzyme. For example, all enzyme batches were purified using a two-step procedure consisting of Ni-NTA affinity chromatography followed by size-exclusion chromatography, protein concentrations were quantified by measuring the absorbance at 280 nm using a calculated extinction coefficient (as was done by Privett et al. and Blomberg et al.), fresh substrate solutions were prepared directly before each kinetic experiment using powder from the same commercial stock, and pH of reaction mixture was adjusted after addition of 10% methanol. Therefore, we expect that our reported kinetic parameters are internally consistent. We have added the following experimental details in the Methods section to clarify our procedures:

“Purified samples were concentrated using Amicon Ultracel-10K centrifugal filter units (EMD Millipore), and quantified by measuring the absorbance at 280 nm and applying Beer-Lambert's law using calculated extinction coefficients (<https://web.expasy.org/protparam/>).”

“Triplicate reactions with varying concentrations of freshly-prepared 5-nitrobenzisoxazole (AstaTech) dissolved in methanol (10% final concentration, pH of reaction mixture adjusted to 7.0 after addition of methanol-solubilized substrate) were initiated by addition of approximately 2 μ M HG3, 50 nM HG3.3b, 10 nM HG3.7/HG3.14, or 5 nM HG3.17/HG4.”

“Initial reaction rates at different substrate concentrations were fit to the Michaelis-Menten or linear equations using GraphPad Prism.”

As far as we can tell, the only significant differences between our protocols and those used by Hilvert and coworkers are the purification protocols (Hilvert and coworkers used Ni-NTA affinity chromatography followed by cation exchange chromatography), the origin of the substrate sample used (we purchased it from AstaTech while Hilvert and coworkers synthesized it in-house), and the use of a 100 mM sodium phosphate buffer instead of 50 mM as used by Hilvert and coworkers. Although the absolute values of $k_{\text{cat}}/K_{\text{M}}$ that we report here are systematically lower than those reported by Hilvert and coworkers, their relative values are similar, and the conclusions drawn from these are not substantially affected.

To calculate $k_{\text{cat}}/K_{\text{M}}$, we used linear regression of rates obtained at low substrate concentrations as described in our response to Reviewer #1's comments. These values are reported in Table 1 (see above).

2) The peptide bond between residues 83 and 84 is described as a mixture of cis and trans configurations for HG3 and HG3.3b but exclusively cis for the more evolved variants (p. 7). While the electron density shown in Fig. 2 for HG3.3b is pretty convincing, the situation is less obvious for HG3. Since the resolution of the latter structure is the lowest of the lot, is modeling the minor trans configuration really justified?

Response: During refinement of the HG3 structures, we observed blobs of unmodelled density around residues 83–84. These blobs aligned perfectly with positions of atoms in the alternate *trans* backbone conformation that we had already modelled in HG3.3b (as the *trans* peptide is the major conformer in this variant). Therefore, we included the *trans* conformer in our HG3 models and performed refinement, which resulted in decreased $R_{\text{work/free}}$ values, as well as

elimination of the positive difference density. These results confirmed our suspicion that both *cis* and *trans* conformations are present at this position in HG3. To demonstrate this, we have added a new supplementary figure (Supplementary Figure 4, see below) where we show the results of refinement of our final models AFTER deletion of the low-occupancy *cis* or *trans* peptide conformation. As shown, the alternate peptide conformation is well-supported by difference density for one or both chains in the asymmetric unit of both HG3 and HG3.3b. A similar effect was also seen for HG3.7 in the unbound state, although the difference features corresponding to the minor conformation were weaker in comparison to HG3 and HG3.3b.

Supplementary Figure 4. Difference density increases around the 83–84 peptide bond after refinement in the absence of minor conformer. The peptide bond between residues 83 and 84 adopts both *cis* and *trans* conformations in HG3, HG3.3b, and HG3.7. To confirm the presence of the alternate peptide, the structure was re-refined in the absence of the minor conformer (transparent sticks), resulting in difference density for one or both chains in the asymmetric unit of HG3 and HG3.3b. A similar effect was also seen for HG3.7 in the unbound state, although the difference features corresponding to the minor conformation were weaker in comparison to HG3 and HG3.3b.

3) In the discussion of active site preorganization, the authors note that Trp44 and Met237 adopt conformations in the apo protein that would prevent productive binding of the TSA. Is the energetic cost of moving these residues out of the way really expected to be high? Along the same lines, Fig 2e seems to suggest that Trp44 is just as disordered in HG3.17 as in its predecessors (p. 9). Have I missed something?

Response: The energy barrier for Trp side-chain conformational changes has been reported to be in the 6–11 kcal/mol range [Houser *PLoS One* 2017, 12, e0189375], which corresponds to the energy of a few hydrogen bonds. Since the active site is expected to be filled with water in the unbound state, it is not surprising that both Trp44 and Met237 are able to sample alternate side-chain conformations, as they are not in a tightly packed environment. As shown in Figure 2e, the side chains of Trp44 or Met237, or both, are observed in all variants to occupy the space where the 6NT transition-state analogue would bind.

Regarding Trp44 in HG3.17, Reviewer #3 is correct: this residue is just as disordered in this high-activity variant as it is in the other variants. However, Met237 is not, as it adopts a single, identical conformer in both the bound and unbound states. Therefore, Met237 is always in the correct, catalytically-productive conformation required to bind 6NT, whereas Trp44 is in a catalytically-productive conformation (i.e. the green one) in ~40% of the molecules in the crystal. In all other variants (with the exception of HG4), either Met237 or Trp44, or both, adopt non-productive side-chain rotamers in the major (magenta) or minor (green) conformation of the unbound state. Thus, there is always one of these residues (Met237 or Trp44) that is in a non-productive conformation in every molecule in the crystal. This is not the case for HG3.17 (or HG4), where ~40% of the molecules in the crystal contain catalytically-productive rotamers for both Trp44 and Met237, making this variant better pre-organized than the other ones (except HG4).

To clarify, we have modified the legend to Figure 2e as follows:

From HG3 to HG3.14, the unbound state is never pre-organized for catalysis as both Trp44 and Met237 adopt conformations that would prevent productive binding of 6NT. In HG3.17 and HG4 however, only Trp44 adopts a non-productive conformation in the unbound state, with an occupancy of 62% or 26%, respectively.

4) The inclusion of two distal mutations in HG4, namely R275A and W276F, weakens the broad claim that distal mutations in HG3.17 contribute little to catalytic efficiency (p. 10). Is the variant lacking these two mutations equally active?

Response: We unfortunately have not tested the HG4 variant that does not contain those two mutations, preventing us from answering this question. As it is highly likely that both mutations contribute to enhanced catalytic activity, we agree with Reviewer #3 that this weakens our broad claim that distal mutations contribute little to catalytic activity. Therefore, we have removed this statement from the manuscript (see our response to Reviewer #1's comment above).

5) For optimization of the HG4 sequence using MD-generated backbone ensembles, how important were the restraints imposed by the HG3 diffraction data? Were controls run simply using MD without the restraints? How did the results compare? Although diffraction data for this system were available, thanks to the authors, that won't generally be the case. Can unrestrained MD simulations be used directly to generate ensembles of relevant backbone templates to improve the design predictions? Some discussion of this issue is important.

Response: The question about how to generate relevant backbone ensembles to increase prediction accuracy is indeed very interesting and important. To evaluate the impact of restraints imposed by the diffraction data when generating an ensemble, we ran a 500-ns unconstrained MD simulation starting from the HG3 (–) 6NT crystal structure (see Methods), and used this trajectory to generate an ensemble of 50 backbone templates. We then used this ensemble to optimize rotamers for the HG4 sequence. As shown in the revised version of Figure 4, use of the ensemble prepared by “unconstrained MD” did improve predictions compared to using the HG3 (–) 6NT crystal structure alone (e.g., Figure 4h vs Figure 4d), but did not outperform predictions made by the ensemble generated using crystallographic restraints (e.g., Figure 4h vs Figure 4f – both the energy and the structure of the best computational model obtained by ensemble refinement are better than those obtained using unconstrained MD).

To rationalize this result, we calculated the diversity (i.e., average root-mean-square backbone coordinate deviation between pairs of ensemble members) and deviation (i.e., average root-mean-square backbone coordinate deviation from the HG4 (+) 6NT crystal structure) of each ensemble, which are metrics that we previously demonstrated to be important for assessing the predictive quality of various backbone ensembles (Davey *Proteins* 2014, 82, 771–784 and Davey *Structure* 2015, 23, 2011–2021). As shown in the new Supplementary Figure 7 (see above), the diversity and deviation of the unconstrained MD ensemble are both much larger than those of the ensemble generated by ensemble refinement. This result is in agreement with our previous observation that the main characteristic that makes an ensemble “on-target” (i.e. a physically valid model of a target protein fold) is its small deviation from the crystal structure (Davey *Proteins* 2014, 82, 771–784).

To describe these results, we have added the following text to the Results section (please also see Supplementary Figure 7 above):

“To evaluate the effect of restraints imposed by the diffraction data, we generated an ensemble using unconstrained molecular dynamics starting from the unbound HG3 crystal structure (Methods), and used it to optimize rotamers for the HG4 sequence. Use of this ensemble resulted in an improved structural model (Figure 4h) compared to the one obtained from the corresponding crystal structure (Figure 4d) that is however less structurally accurate and stable than the one obtained from ensemble refinement (Figure 4f). The better predictive ability of the HG3-derived ensembles prepared using crystallographic restraints likely results from their lower deviation from the HG4 crystal structure (≈ 0.4 Å, Supplementary Figure 7), which we previously showed to be necessary for an ensemble to represent a physically valid model of the target protein fold.²⁴ Overall, these results suggest that computational enzyme design with a crystallographically-derived backbone ensemble derived from a low activity enzyme could obviate the need for directed evolution by allowing catalytically-competent sub-states to be sampled during the design procedure.”

Although our results suggest that an ensemble generated from an unconstrained MD simulation performs worse than one generated from ensemble refinement, we cannot exclude the possibility that an alternate ensemble generated by a different unrestrained MD protocol (e.g., longer simulation time, alternate force field and simulation conditions, etc.) would not perform equally or better than the ensembles restrained by the diffraction data. Additional experiments are required to answer this question, which we are currently investigating.

Reviewer #3 is correct in that at the beginning of an enzyme design project, the crystal structure and density map for an active *de-novo*-designed variant would not be available. Thus, design can only be performed starting from data available in the PDB, which would likely be for proteins

devoid of the target catalytic activity (for example, the 1GOR template used by Privett et al. to design HG2). For this reason, we have also generated an ensemble from the 1GOR diffraction data using ensemble refinement, and used it to design the HG4 sequence. The results shown in Figure 4 and Supplementary Figure 6 demonstrate that use of this ensemble slightly improves predictions relative to using the 1GOR crystal structure (e.g., energy is lowered by ~10 kcal/mol but transition-state binding pose remains inaccurate). However, predictions still remain worse than those obtained using ensembles generated from the HG3 (+) 6NT or HG3 (-) 6NT data. The lower performance of the 1GOR ensemble results from the inability of the residue 83 backbone to sample a conformation required to bind the transition state in the crystallographic binding pose (see Supplementary Figure 7 above). These results support our proposal that an iterative approach to computational enzyme design that utilizes a backbone ensemble generated from experimental structural data obtained for an initial, low-activity enzyme could circumvent the need for directed evolution, which we are now investigating.

6) Producing more active computational enzymes is an important goal in the design community which could minimize or eliminate the need for tedious optimization by directed evolution. Although the authors have shown that they can computationally design a *de novo* Kemp eliminase with native-like catalytic efficiencies when they already know what an active enzyme looks like, it is not yet clear to what extent their findings will translate to other scaffolds and other reactions if this information is not available ahead of time.

Response: Reviewer #3 is absolutely correct that it is not yet clear how to use this method to design other *de novo* enzymes. Our current hypothesis, as proposed in the discussion section of this manuscript, is that an iterative approach to computational enzyme design in which an additional round of design that utilizes a backbone ensemble generated from experimental structural data obtained for an initial, low-activity enzyme, will help to circumvent the need for directed evolution and yield high-activity *de novo* enzymes. We are currently testing this hypothesis to design new *de novo* enzymes. However, the design and validation of new enzymes is time-intensive, and very much beyond the scope of the present manuscript, the purpose of which was to evaluate changes along the evolutionary trajectory to the conformational ensemble of a designed enzyme, and describe an ensemble-based enzyme design approach.

Additional Changes:

While addressing reviewer comments, we revisited our enzyme design scripts and found minor variations between the protocols used to design HG4 on different templates. Specifically, for some simulations we allowed 2 additional residues to sample alternate rotamers (but without varying their amino-acid identity). Inclusion of these additional rotamers resulted in small energy differences (≈ 2 kcal/mol or less) but did not substantially impact the structures themselves, and therefore our conclusions are not affected. Nevertheless, to be consistent, we have updated Figure 4 and Supplementary Figure 6 to include results obtained from simulations performed using identical protocols.

REVIEWER COMMENTS

Reviewer #1 (Remarks to the Author):

I think most of the changes are fine and the paper is improved - it's a very nice piece of work. However, I'm still confused about the enzyme kinetics... The authors state:

Catalytic efficiencies (k_{cat}/K_M) were obtained by linear regression of six or nine individual replicates of rates measured at the three or four lowest substrate concentrations from two or three independent protein batches. Errors of curve fitting are provided.

Linear regression could be used to estimate the k_{cat}/K_M - but you would need to make a double reciprocal lineweaver burk plot - from which you could obtain the K_M/V_{max} from the slope. In which case data shouldn't be discarded.

But I am confused by what the authors have described - "curve fitting" is non-linear regression. Even then I am unclear as to why only some data points were used (if making a reciprocal plot why not use all data). Estimating error from the "curve" (linear?) is also confusing to me. Typically for non linear regression, you can estimate the error in k_{cat} and K_M using graphing programs (error for k_{cat}/K_M is not usually reported). If you wanted to report the error for k_{cat}/K_M you would need to calculate the error of the ratio and explain how this was done. If the authors have just truncated the data to the linear part of the curve, then used the standard non-linear curve fitting program to fit the M-M equation in graphpad (to a graph of v vs $[s]$), I'm not sure this is appropriate? If they have used an approach like that and they think it is suitable then it would be best to cite the paper showing it is a robust method for estimating k_{cat}/K_M (apologies for not being aware of it).

If combining errors of k_{cat} and K_M this paper is useful:

<https://onlinelibrary.wiley.com/doi/pdf/10.1002/9781118540398.app5>

Maybe there is a simple way this is done that I am unaware of (I've looked up a lot of textbooks and unless it is a double reciprocal plot I can't work out how this was done with linear "curve fitting"), but I think it needs to be described more clearly, in terms of equations, curve (?) fitting, error calculation, etc., i.e. the substrate vs velocity graphs in the SI are good, but these were not used to derive the data in the table - whatever was done to generate that data (graphs, linear regression) needs to be presented.

I should qualify this by saying I think the estimates seem roughly fine (errors seem too low but not clear how they are estimated) but it is better that the enzyme kinetics are described clearly and performed correctly for anybody following this work.

Reviewer #3 (Remarks to the Author):

The revised manuscript does a good job in addressing the points raised in my original critique. The systematically lower values of the steady-state parameters compared to those previously reported is the only remaining concern. The authors note that they took care to minimize possible sources of error, which is laudable, and their results are internally consistent. Although the deviations may not alter the conclusions of the study, one nevertheless wonders about their origin. Curiously, were the kinetic measurements performed individually in cuvettes in a spectrophotometer or in microtiter plates in a plate reader? If the latter, uncertainty about the actual pathlength in the individual wells may have been a contributing factor. Regardless, this is a fine study and merits publication as a Nature Communication.

Response to Reviewers and List of Changes

Reviewer #1

I think most of the changes are fine and the paper is improved - it's a very nice piece of work. However, I'm still confused about the enzyme kinetics... The authors state:

Catalytic efficiencies (k_{cat}/K_M) were obtained by linear regression of six or nine individual replicates of rates measured at the three or four lowest substrate concentrations from two or three independent protein batches. Errors of curve fitting are provided.

Linear regression could be used to estimate the k_{cat}/K_M - but you would need to make a double reciprocal lineweaver burk plot - from which you could obtain the K_M/V_{max} from the slope. In which case data shouldn't be discarded.

But I am confused by what the authors have described - "curve fitting" is non-linear regression. Even then I am unclear as to why only some data points were used (if making a reciprocal plot why not use all data). Estimating error from the "curve" (linear?) is also confusing to me. Typically for non linear regression, you can estimate the error in k_{cat} and K_M using graphing programs (error for k_{cat}/K_M is not usually reported). If you wanted to report the error for k_{cat}/K_M you would need to calculate the error of the ratio and explain how this was done. If the authors have just truncated the data to the linear part of the curve, then used the standard non-linear curve fitting program to fit the M-M equation in graphpad (to a graph of v vs $[S]$), I'm not sure this is appropriate? If they have used an approach like that and they think it is suitable then it would be best to cite the paper showing it is a robust method for estimating k_{cat}/K_M (apologies for not being aware of it).

If combining errors of k_{cat} and K_M this paper is useful:

<https://onlinelibrary.wiley.com/doi/pdf/10.1002/9781118540398.app5>

Maybe there is a simple way this is done that I am unaware of (I've looked up a lot of textbooks and unless it is a double reciprocal plot I can't work out how this was done with linear "curve fitting"), but I think it need to be described more clearly, in terms of equations, curve (?) fitting, error calculation, etc., i.e. the substrate vs velocity graphs in the SI are good, but these were not used to derive the data in the table - whatever was done to generate that data (graphs, linear regression) needs to be presented.

I should qualify this by saying I think the estimates seems roughly fine (errors seem too low but not clear how they are estimated) but it is better that the enzyme kinetics are described clearly and performed correctly for anybody following this work.

Response: I apologize if there is still confusion, which I believe stems from our misuse of "curve fitting" to describe "linear fitting". I will try to clarify further below how we determined k_{cat}/K_M for every enzyme.

1. We purified each enzyme 2 or 3 separate times (ie different protein batches purified at different times).
2. With each batch, we performed triplicate rate measurements at 7 or more substrate concentrations between 0 and 2 mM using the assay described in methods.

3. We averaged all individual rates obtained at the same substrate concentration over all batches of the same enzyme. This gave 6 or 9 rate replicates for enzymes purified twice or thrice, respectively. Michaelis-Menten plots are shown in Supplementary Figure 2 (no data was omitted). Even though we could fit the data to the Michaelis-Menten equation using GraphPad, we did not use the v_{\max} and K_M values obtained from these fits to calculate k_{cat}/K_M because the fits gave very large errors on K_M and v_{\max} . This is due to the fact that we could not saturate the enzyme at the highest substrate concentration tested (2 mM) due to the solubility limit of 5-nitrobenzisoazole.
4. Since we could not saturate the enzymes within the substrate's solubility limit, we used linear regression (ie, $y = mx$, where $y = v_0$, $m = v_{\max}/K_M$, $x = [S]$) on the rates obtained using the 3 or 4 lowest substrate concentrations (ie those that gave a linear regression with R^2 close to 1.0) to calculate k_{cat}/K_M . At low substrate concentrations (ie when $[S] \ll K_M$), k_{cat}/K_M is directly proportional to the slope of the linear part of the hyperbolic Michaelis-Menten plot. This procedure is exactly what Reviewer #3 asked us to do in their Comment #1 (*"measure very accurate kcat/Km values by fitting a line through the data obtained at substrate concentrations well below Km"*).
5. Errors reported on Table 1 are the errors of the linear fit, ie the absolute measure of the typical distance that each data point falls from the regression line. This is what Reviewer #1 had asked us to report, except that it is for a linear instead of a non-linear regression (*"The more common way to do this would be to plot all the data on a single graph, fit a curve to the data and use the error of the curve fitting to estimate the error"*).

Furthermore, we did NOT use the Lineweaver-Burke double reciprocal analysis as this would not solve the issue caused by the fact that we could not saturate the enzyme with substrate (ie we would still obtain large errors on v_{\max} and K_M).

In a nutshell: data were fitted to the linear portion of the Michaelis-Menten model ($v_0 = (k_{\text{cat}}/K_M)[E_0][S]$), and k_{cat}/K_M was deduced from the slope.

To clarify, we have modified the (1) Supplementary Figure 2 caption, (2) methods, and (3) Table 1 footnotes:

- (1) **Supplementary Figure 2. Steady-state kinetics.** Michaelis–Menten plots of normalized initial rates as a function of 5-nitrobenzisoazole (5NBZ) concentrations are shown. Data represent the average of six or nine individual replicate measurements from two or three independent protein batches, with error bars indicating the SEM. Saturation was not achieved for any enzyme at the substrate's solubility limit (2 mM). Therefore, only k_{cat}/K_M values are reported on Table 1, and these values were calculated using linear regression of rates measured at the three or four lowest substrate concentrations. Under these conditions, $[S] \ll K_M$, the Michaelis-Menten equation becomes $v_0 = (k_{\text{cat}}/K_M)[E_0][S]$.
- (2) Data were fitted to the linear portion of the Michaelis-Menten model ($v_0 = (k_{\text{cat}}/K_M)[E_0][S]$), and k_{cat}/K_M was deduced from the slope.
- (3) Catalytic efficiencies (k_{cat}/K_M) were calculated from the slope of the linear portion ($[S] \ll K_M$) of the Michaelis-Menten model ($v_0 = (k_{\text{cat}}/K_M)[E_0][S]$). Errors of linear regression

fitting, which represent the absolute measure of the typical distance that each data point falls from the regression line, are provided.

Reviewer #3

The revised manuscript does a good job in addressing the points raised in my original critique. The systematically lower values of the steady-state parameters compared to those previously reported is the only remaining concerning. The authors note that they took care to minimize possible sources of error, which is laudable, and their results are internally consistent. Although the deviations may not alter the conclusions of the study, one nevertheless wonders about their origin. Curiously, were the kinetic measurements performed individually in cuvettes in a spectrophotometer or in microtiter plates in a plate reader? If the latter, uncertainty about the actual pathlength in the individual wells may have been a contributing factor. Regardless, this is a fine study and merits publication as a Nature Communication.

Response: We performed all of our enzyme assays in 96-well microtiter plates using a SpectraMax 384Plus plate reader. Path lengths for each well were calculated ratiometrically using the equation below:

$$\text{Pathlength} = (A_{975}(\text{well}) - A_{900}(\text{well})) / (A_{975}(\text{cuvette}) - A_{900}(\text{cuvette})) * 1 \text{ cm}$$

where A975 and A900 indicate the absorbance at 975 and 900 nm (27 °C) of our reaction solution (100 mM sodium phosphate buffer (pH 7.0) supplemented with 100 mM sodium chloride and 10% methanol) in a microplate well or 1-cm cuvette.

In our experience, this procedure results in $\leq 5\%$ error compared to using a 1-cm cuvette. We therefore do not expect that differences between the steady-state parameters reported here and those reported previously [Blomberg et al., *Nature* 2013] are primarily caused by the fact that we used microplates instead of cuvettes to perform our enzyme assays, although it certainly can contribute.

To clarify our procedures, we have modified the description of our enzyme assay, and provided a reference for the ratiometric procedure described above (new Reference #31):

Steady-state kinetics. All assays were carried out at 27 °C in 100 mM sodium phosphate buffer (pH 7.0) supplemented with 100 mM sodium chloride. Triplicate 200- μL reactions with varying concentrations of freshly-prepared 5-nitrobenzisoazole (AstaTech) dissolved in methanol (10% final concentration, pH of reaction mixture adjusted to 7.0 after addition of methanol-solubilized substrate) were initiated by addition of approximately 2 μM HG3, 50 nM HG3.3b, 10 nM HG3.7/HG3.14, or 5 nM HG3.17/HG4. Product formation was monitored spectrophotometrically at 380 nm ($\epsilon = 15,800 \text{ M}^{-1} \text{ cm}^{-1}$)⁵ in individual wells of 96-well plates (Greiner Bio-One) using a SpectraMax 384Plus plate reader (Molecular Devices). Path lengths for each well were calculated

ratiometrically using the difference in absorbance of 100 mM sodium phosphate buffer (pH 7.0) supplemented with 100 mM sodium chloride and 10% methanol at 900 and 975 nm (27 °C).³¹ Linear phases of the kinetic traces were used to measure initial reaction rates. Data were fitted to the linear portion of the Michaelis-Menten model ($v_0 = (k_{\text{cat}}/K_{\text{M}})[E_0][S]$), and $k_{\text{cat}}/K_{\text{M}}$ was deduced from the slope.